# The influence of human activities on streamflow reductions during the megadrought in Central Chile

Nicolás Álamos[1,2,3], Camila Alvarez-Garreton[1], Ariel Muñoz[1,3,4], Álvaro González-Reyes[2,5,6,7]

[1]Center for Climate and Resilience Research (CR2, FONDAP 1522A0001), Santiago, Chile, Chile
[2]Instituto de Ciencias de la Tierra ICT, Facultad de Ciencias, Universidad Austral de Chile, Chile
[3]Centro de Acción Climática, Pontificia Universidad Católica de Valparaíso, Chile
[4] Laboratorio de Dendrocronología y Estudios Ambientales, Instituto de Geografía, Pontificia Universidad Católica de Valparaíso, Chile
[5] Centro de Humedales río Cruces CEHUM, Universidad Austral de Chile, Chile
[6]Laboratorio de Dendrocronología y Cambio Global, Universidad Austral de Chile, Valdivia, Chile
[7]Centro de Investigación: Dinámica de Ecosistemas Marinos de Altas Latitudes - IDEAL, Chile
*Correspondence to*: Camila Alvarez Garreton (calvarezgarreton@gmail.com)

**Abstract.** Since 2010, central Chile has experienced a protracted megadrought with annual precipitation deficits ranging from 25% to 70%. An intensification of drought propagation has been attributed to the effect of cumulative precipitation deficits linked to catchment memory. Yet, the influence of water extractions on drought intensification is still unclear. Our study assesses climate and water use effects on streamflow reductions during a high human influence period (1988-2020) in four major agricultural basins. We performed this attribution by contrasting observed streamflow (driven by climatic and water use) with near-natural streamflow simulations (driven mainly by climate) representing what would have occurred without water extractions. Near-natural streamflow estimations were obtained from rainfall-runoff models trained over a reference period with low human intervention (1960-1988). Annual and seasonal streamflow reductions were examined before and after the megadrought onset, and hydrological drought events were characterized for the complete evaluation period in terms of their frequency, duration and intensity.

Our results show that before the megadrought onset (1988-2009), the mean annual deficits in observed streamflow ranged between 2 to 20% across the study basins, and that 81 to 100% of those deficits were explained by water extractions. During the megadrought (2010-2020), the mean annual deficits in observed streamflow were 47 to 76 % among the basins. During this time, the relative contribution of precipitation deficits on streamflow reduction increased while the contribution of water extractions decreased, accounting for 27 to 51% of the streamflow reduction. Regarding drought events during the complete evaluation period, we show that human activities have amplified drought propagation, with almost double the intensity of hydrological

droughts in some basins, compared to those expected by precipitation deficits only. We conclude that while the primary cause of streamflow reductions during the megadrought has been the lack of precipitation, water uses have not diminished during this time, causing an exacerbation of the hydrological drought conditions and aggravating their impacts on water accessibility in rural communities and natural ecosystems.

## 1 Introduction

The fluxes of the water cycle vary and change in time and space, as well as the anthropic activities affecting those fluxes, leading to a co-evolving hydrosocial cycle (Linton and Budds, 2014; Budds, 2012) that defines the state of the hydrological system (Van Loon et al., 2016). Observational evidence in different regions indicates that hydrological cycles are being affected by climate change and human activities. Climate change has led to changes in precipitation patterns worldwide (Fleig et al., 2010; Kingston et al., 2015), while human activities have altered the spatiotemporal distribution of water resources (Van Loon et al., 2022). This can lead to water scarcity problems, particularly when precipitation deficits occur in regions that concentrate water consumption requirements.

The alterations in the water cycle may also affect the occurrence of droughts, which are defined as a deficit of water relative to normal conditions and can be identified in different components of the hydrological cycle. While meteorological droughts (precipitation deficits) are mainly controlled by regional climate, hydrological droughts (streamflow, and groundwater deficits) are also influenced by catchment characteristics and water uses. In this way, under similar meteorological conditions, the severity of hydrological droughts and their impacts on society can vary significantly within the territory (Van Lanen et al., 2013).

Most drought analyses consider climate variability as a main driver of drought, however, increasing focus has been given to assessing the compounding effects of climate variability and human activities on water resources and drought propagation (Van Loon et al., 2016; Wanders and Wada, 2015; Zhao et al., 2014). Anthropic activities, such as irrigation, urbanization, land use changes, and water infrastructure (e.g., reservoirs or water transfer channels) affect runoff mechanisms (Huang et al., 2016) and can lead to a higher frequency of hydrological droughts (Alvarez-Garreton et al., 2021; Ward et al., 2020). An example of this is the Yellow river basin in China, where despite no significant rainfall deficits have occurred in recent years, a hydrological drought with historical minimum streamflow levels is being observed, which has been mainly driven by anthropic activities in the basin (Huang et al., 2016; Kong et al., 2016; Li et al., 2019; Liu et al., 2016; Zhao et al., 2014).

Advancing our understanding of hydrological droughts as a complex process depending on the interaction between climatic, biophysical, and anthropic drivers is critical to assessing catchment's vulnerability to droughts, mitigating their occurrence, and designing adaptation plans. While all these drivers influence the propagation and impacts of droughts, water management plans mainly influence human activities and their local disturbances to the hydrological cycle. Therefore, it is critical to address the scientific challenge of understanding the influence of human activities on the hydrological cycle and quantifying their impacts.

To address this challenge, in this paper we focus on central Chile (29°-35°S; Fig. 1), a region where the signal of anthropic climate change is leading to an increase in mean temperature, increasing of heatwaves events, and a sustained decrease in precipitation (Boisier et al., 2018; Bozkurt et al., 2017; Garreaud et al., 2017, 2020; González-Reyes et al., 2023). The drying trend has led to the so-called megadrought, affecting the country since 2010, with annual precipitation deficits ranging between 25% and 70% (Garreaud et al., 2020, 2017). This meteorological drought in central Chile has propagated across the terrestrial system, leading to hydrological droughts and water scarcity problems that vary across the territory (Alvarez-Garreton et al., 2021; Duran-Llacer et al., 2020; Muñoz et al., 2020; Barría et al., 2021b).

In the Petorca river basin, located in the Valparaiso region in central Chile, Muñoz et al. (2020) found that during the megadrought, streamflow and water bodies from the upper parts of the basin were less affected than the mid and low areas of the valley, where most of the agriculture is located. However, the authors did not make a formal attribution about the role of water consumption and climate on streamflow reduction. Another study was conducted on the Aculeo Lake, a natural reservoir in central Chile that dried up during the ongoing megadrought. Barría et al (2021b) performed an attribution exercise by using the Water Evaluation and Planning System (WEAP) hydrological model and concluded that climate was the primary factor explaining the lake's drying, while water demand has remained stable over the past few decades. Another study reported that basins with larger human intervention within this region exhibited lower runoff sensitivities to precipitation compared to less disturbed ones (Alvarez-Garreton et al., 2018). In that study, the authors attributed this phenomenon to the alteration of runoff generation mechanisms associated with water withdrawals and reservoirs. Furthermore, higher than expected streamflow reductions during the megadrought have also been observed in near-natural basins. Alvarez-Garreton et al. (2021) reported the effects of catchment memory in snow-dominated catchments in Central Chile, where the accumulation of the persistent precipitation deficits led to less streamflow than expected from observations during previous single-year meteorological droughts. These studies have advanced our understanding about the role of catchments and anthropic characteristics in the

megadrought's propagation, however, further studies are still required to robustly assess the impacts of human activities on
streamflow reduction and drought conditions in the major basins of central Chile.
In this article, we quantify the relative effects of climate and water extractions on streamflow reduction in four major agricultural
basins in central Chile. We analyse a period with high human influence within the study basins (1988-2020), and assess how the
relative effects of climate and water extractions change before and after the megadrought onset. Additionally, we assess the
influence of water extractions on the intensity, frequency, and duration of hydrological droughts for the complete evaluation
period. To achieve this, we follow the approach proposed by Van Loon et al. (2022) and compare streamflow observations with
a near-natural simulated streamflow representing the discharge that would have occurred without human influences.
Hydrological droughts are identified by streamflow deficit using a threshold determined from the near- natural scenario, allowing
for better identification of human impacts (Van Loon, 2016).
**2 Methods and data**
**2.1 Study area**
The study was conducted in four major basins located between 29° and 33°S (Fig. 1): The Elqui, Limarí, and Choapa basins in
the Coquimbo region, and the Aconcagua basin in the Valparaíso region. These basins fall within semi-arid (Coquimbo region)
and Mediterranean (Valparaiso region) climate zones, which are particularly vulnerable to droughts due to the majority of annual
precipitation occurring during the winter season concentrated on a few storm events (Garreaud et al., 2017).
All catchments feature a snow-rain-fed hydrologic regime. The Aconcagua basin also has a large glacier area (192 km$^2$) that
contributes to streamflow, especially during dry summers (Crespo et al., 2020). The study basins have experienced precipitation
deficits of 25-70% and streamflow deficits of up to 70% during the megadrought that has affected the region since 2010 (Alvarez-
Garreton et al., 2021; Garreaud et al., 2020, 2017).
According to the data provided by the water security platform from the Center for Climate and Resilience Research
(www.seguridadhidrica.cl), agriculture is the primary productive sector and the main consumer of water resources within these
basins. Agricultural land cover areas of 152 km$^2$ (total catchment area of 9800 km$^2$), 605 km$^2$ (total catchment area of 11800
km$^2$), 313 km$^2$ (total catchment area of 8124 km$^2$), and 582 km$^2$ (total catchment area of 7200 km$^2$), and their annual water
consumption at present corresponds to 3.25 m³/s, 14.3 m³/s, 6.48 m³/s, and 15.72 m³/s, in the Elqui, Limarí, Choapa, and
Aconcagua basins, respectively. Avocado and table vine species are the main consumers in the Aconcagua basin, while the
Limarí basin has a higher demand from permanent forage species, table vine, and citrus plantations.

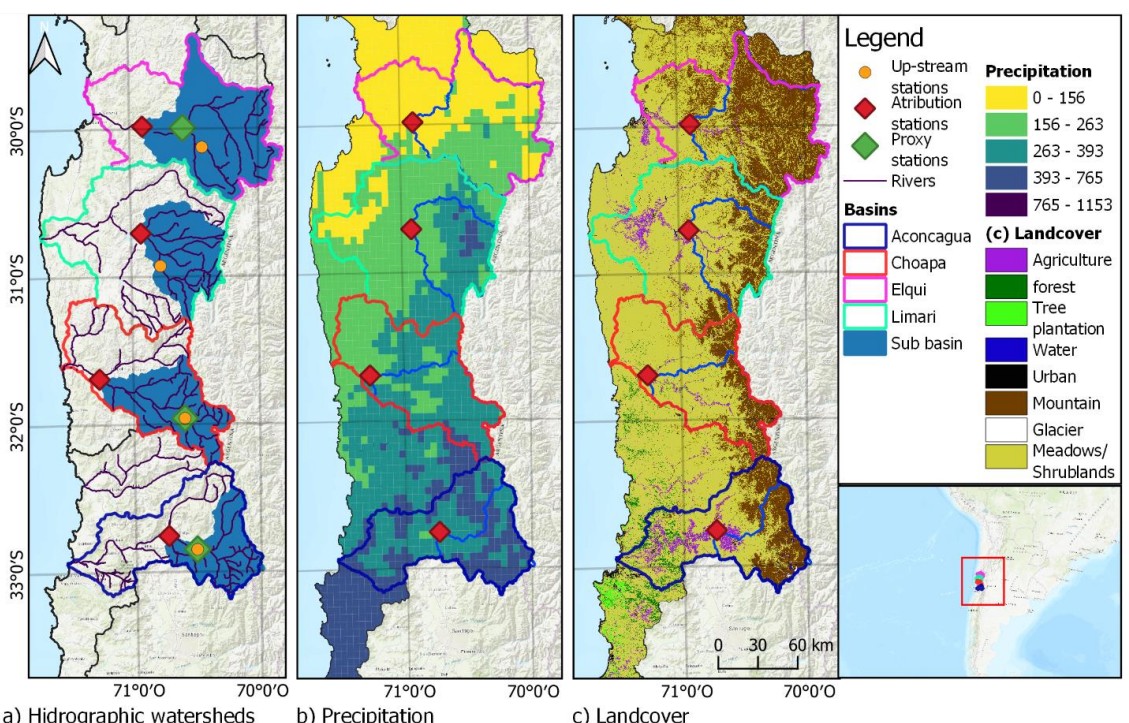

a) Hidrographic watersheds   b) Precipitation   c) Landcover


**Figure 1. Panel a) shows the four main basins of the study area and the streamflow gauges used for the analyses. The red diamonds**
**indicate the stations used to characterise each basin; the green diamonds are the gauges used as predictors for filling in monthly**
**streamflow data (note that in the Limarí catchment, no gap-filling process was made, leading to the absence of a predictor gauge**
**station; Sect. 2.2); and the orange circles are the up-stream stations used in the rainfall-runoff ratio analysis (Sect. 2.3). The basin area**
**covered by the red diamond gauge is painted blue. Panel b) presents the mean annual precipitation (mm/yr) from the CR2MET dataset**
**for the period 1980-2010. Panel c) shows the gridded land cover dataset from Zhao et al. (2016). Base map source: Esri, 2017.**
**2.2 Data**
Catchment boundaries and times series of total monthly streamflow normalized by catchment area (in mm/month) were obtained
from the CAMELS-CL dataset (Alvarez-Garreton et al., 2018; available at: https://camels.cr2.cl/) for the period April 1960 –

March 2020. Total monthly precipitation for the same period was obtained from the CR2MET dataset version 2.5 at a 5 x 5 km grid resolution (Boisier, 2023) and averaged across the basin boundaries. Catchment-scale monthly evapotranspiration (ET) was computed based on the ECMWF surface re-analysis ERA5-Land dataset, available at a horizontal resolution of 10 km (Muñoz-Sabater et al., 2021) from April 1960 to March 2020. For each study basin, we selected the most downstream streamflow gauge station having more than 80% of streamflow records for the 1960-2020 period (see Fig. 1). Gaps in monthly streamflow of downstream gauges (red diamonds in Fig. 1a) were filled based on linear regression models, using the basin's precipitation and the streamflow of an upstream gauge with a strong correlation with the considered station (green diamonds in Fig. 1a) as predictors. The linear regressions resulted in coefficients of determination larger than 0.8 in Elqui, Choapa, and Aconcagua basins.

Streamflow and basin-averaged precipitation and ET were computed for hydrological years (April to March in Chile) and for wet and dry seasons. The wet season is defined from April to August, while the dry season corresponds to the months between September and March. Annual (seasonal) streamflow values were computed when the 12 (6) months had valid data.

To account for human intervention within the basins, we analysed annual water uses from industry, energy, mining, livestock, drinking water sectors, as well as water evaporation from lakes and reservoirs for the period 1960-2020 obtained from the water security platform from the Center for Climate and Resilience Research (www.seguridadhidrica.cl). All variables with a different spatial resolution than the basin (whether gridded or administrative units) were calculated for the basin considering the weighted average of the variable within the basin surface.

**2.3 Near-natural streamflow modelling and attribution exercise**

The attribution exercise to quantify the climatic and human contributions on streamflow reductions is schematized in Fig. 2. Near-natural streamflow simulations were obtained by rainfall-runoff statistical models trained in a period when anthropic activities had low water consumption (Sharifi et al., 2021; Zhao et al., 2014).

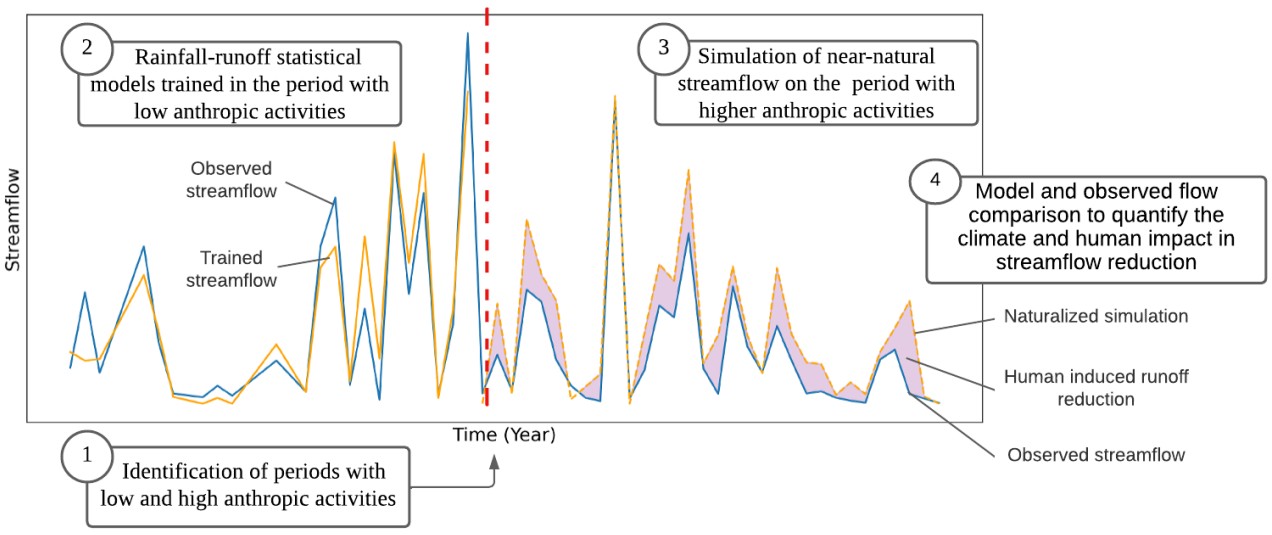

138

**Figure 2. Flowchart of the steps to quantify the human contribution to streamflow reduction based on comparing a near-natural simulated streamflow with the observed streamflow on a period of high anthropic activities.**

### 2.3.1 Selection of low-influence reference periods

For each basin, we identified low human intervention periods based on the regime shifts of streamflow, precipitation, and water uses (Sect. 2.2). The non-parametric Buishand break point test (Buishand, 1982) was applied to identify these shifts. Buishand is a statistical homogeneity test method that checks if two (or more) datasets come from the same distribution. In this way, the test can detect breakpoints where the distribution of a dataset changes. We applied the Buishand test to each time series during the 1960-2020 periods. To identify multiple breakpoints, we iterated the test in the sub-periods before and after the previous breakpoint until no breakpoints with a significance level at p-value < 0.05. For the Buishand test, we used the pyHomogeneity Python library (Shourov, 2020).

In order to select periods with minimal human activities, it is important to identify breakpoints in the streamflow time series that are not primarily explained by climate shifts. To account for this, we selected a unique training period across basins based on the identification of concurrent breaking points in both streamflow and human activities time series, while ensuring the absence of discernible precipitation shifts. We analysed multiple variables instead of using only water use data to achieve a more robust

selection of the training period. This reduces the effects of inter-basin water transfers and land cover changes, which may obscure
the ability of water use data to accurately capture the magnitude of anthropogenic intervention in the basins.
To ensure that the chosen period of analysis is not dependent on the specific statistical test employed, we conducted a sensitivity
analysis using the Sequential T-test Analysis of Regime Shifts (STARS) at a monthly time scale for both precipitation and
streamflow time series (Rodionov, 2004). The STARS V6.3 Excel macro application, available at
https://sites.google.com/view/regime-shift-test was utilized to perform the STARS test.
**2.3.2 Climate and human contribution to streamflow reduction**
Assuming that the effects of climate and local human activities on streamflow generation are independent, the observed
streamflow ($Q_{obs}$) can be disaggregated as follows (Kong et al., 2016):
$$Q_{obs} = Q_{nn} + \Delta Q_{human} \quad (1)$$
Where $Q_{nn}$ corresponds to a climatic-induced streamflow, referred as near-natural streamflow in this paper, and $\Delta Q_{human}$ is the
human-induced effect on streamflow. In this study, near-natural streamflow in Eq.1 is estimated from linear rainfall-runoff
regressions trained during the low-influence reference period defined in Sect. 2.3.1. To account for pluvial and snowmelt runoff
generation processes, we implemented seasonal rainfall-runoff models considering the total streamflow and rainfall in the six-
month periods defined in Sect 2.2 as dependent and independent variables, respectively. In several snow-dominated basins in
central Chile, the winter flows continue to be fed by the snow melt from the previous hydrological year, especially when the
previous year was wetter than normal (Alvarez-Garreton et al., 2021). Given this, winter flow models include winter precipitation
from the previous year. The models representing near-natural summer ($\hat{Q}_{summer}$) and near-natural winter streamflow
($\hat{Q}_{winter}$) were defined for year $t$ as follows:
$$\hat{Q}_{summer}(t) = a_0 + a_1 P_{winter}(t) \quad (2)$$
$$\hat{Q}_{winter}(t) = b_0 + b_1 P_{winter}(t) + b_2 P_{winter}(t-1) \quad (3)$$
The coefficients in Eq. 2 and 3 were obtained by least square errors method during the training period. Based on this, the human
influence during the evaluation (high-influence) period was obtained as:

$$\Delta Q_{human} = Q_{obs} - \hat{Q}_{nn} \pm \varepsilon \quad (4)$$

where $\widehat{Q_{nn}}$ is the simulated near-natural streamflow (seasonal concatenation of Eq. 2 and 3) and $\varepsilon$ represents the uncertainty from the regression model parameters. The attribution exercises were performed by applying Eq. 4 during the evaluation period. In the results of the attribution exercise (Sect 3.3; Fig 7), hydroclimatic variables are depicted as anomalies computed as the percentage difference from their mean values during the reference period (1960-1988). Noteworthy that multiple regression equations with different functional forms (including a Box-Cox transformation to the seasonal and annual streamflow to account for potential non linearities between precipitation and streamflow) and variables (such as evapotranspiration and temperature) were tested for representing near-natural streamflow during the reference period (see Appendix A). The linear rainfall-runoff regressions from equations (2) and (3) were those with a higher $r^2$, and all variables were statistically significant at a p-value of 0.05.

It should be noted that the near-natural streamflow estimations from Eq. 2 and 3 assume a stationary rainfall-runoff relationship. However, recent evidence in this region has shown that under protracted drought conditions, a non-stationary catchment response modulated by catchment memory can emerge, resulting in larger streamflow reductions than those expected from single-year precipitation deficits (Alvarez-Garreton et al., 2021). This evidence corresponds to the headwater near-natural basins located upstream of the human influenced basins selected in this study. To assess whether our analyses over the complete basins are potentially biased by non-stationary catchment responses, we compared the rainfall-runoff ratios (mean annual observed streamflow normalised by mean annual precipitation) during the evaluation period before (1988-2010) and after the megadrought onset (2010-2020), in both the upper and lower sections of each basin. These sections were defined by the streamflow gauges highlighted in orange circles and red diamonds in Fig. 1, respectively.

**2.4 Hydrological drought events characterisation**

To quantify the impact of human activities on hydrological droughts (schematised in Fig. 3), we compared the characteristics of the observed and the near-natural streamflow deficits during drought events, including their frequency (number of drought events), duration (average, maximum and total seasons), and intensity (i.e., deficit of volume) across the evaluation period. In this way, we assessed the influence of human activities over observed hydrological droughts by calculating the relative difference in each drought characteristic (DC) in the observed and near natural scenario. To keep consistency with the attribution methodology (Sect. 2.3), drought events were characterised at a seasonal scale, as indicated in Eq. 5.

202
$$DC_{human} = \frac{DC_{obs} - DC_{nn}}{DDC_{obs}} * 100 \quad (5)$$

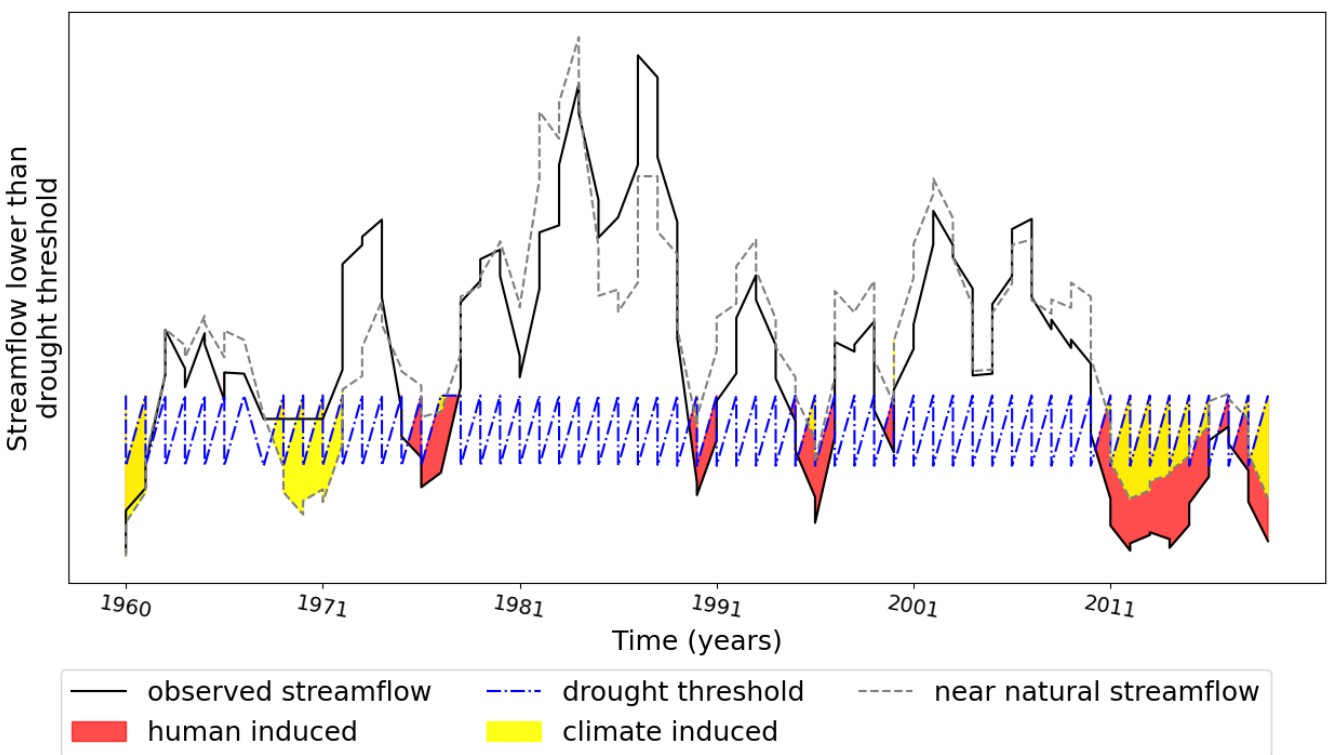

203

**Figure 3. Example of drought periods with annual streamflow lower than a threshold. Three types of droughts are identified: climate-induced droughts, when near-natural streamflow simulations are below the threshold; human-induced droughts, where only observations are below the threshold; and human and natural induced, where both observations and near-natural estimations are below the threshold (adapted from Van Loon et al., 2016).**

To identify the hydrological drought events, we adopted a threshold approach, which defines drought events when the streamflow
is below a specific percentile of the flow duration curve. For daily or monthly time series, a recommended threshold falls between
the 70th and 90th percentile (Rangecroft et al., 2019; Van Loon et al., 2016; Van Loon, 2015). In this study, we adopted the 70th
percentile of the seasonal streamflow series. This lowest threshold allows for the selection of more drought events, which makes
statistical analysis more robust. The threshold can be fixed or variable; we used the variable threshold to incorporate seasonality
into the drought selection (Rangecroft et al., 2019; Van Loon et al., 2019).
To allow for a strict assessment of human influence on hydrological drought, the selected threshold should not account for human
activities (Rangecroft et al., 2019. To achieve this, we defined the 70th percentile threshold based on the entire period of records
(1960-2020), but considering a naturalized regime provided by the near-natural simulated streamflow time series.  It should be
noted that if the observed streamflow for the complete period were considered, human activities would be included. On the other
hand, if only the training low-influence periods were used to calculate the threshold, the climate variability and drying trend of
the complete period would not be represented by the threshold.
**3 Results**
**3.1 Low-influence reference period**
The series of annual streamflow, precipitation, total evapotranspiration (ET), and runoff coefficients (runoff normalised by
precipitation) are shown in Fig. 4. The Buishand test resulted in significant change points only in streamflow and ET. Three
change points were detected in all basins, the first between the years 1977-1978, the second one in 1988, and the last one between
years 1998-2010 years for the streamflow in all basins (Fig. 4), while a single change point was detected in 1973-1975 for ET
in all basins except Aconcagua (Fig. 4d). The STARS test detected similar three change points in streamflow in 1977-1981,
1988, and 2010, with the 1988 breakpoint presenting the higher R-shift index value.
The streamflow breakpoint of 1977-1978 was disregarded since it is mainly due to climatic drivers, as indicated by the single
ET breakpoint during that period. We can relate this to the great Pacific shift and the warm cycle of the Pacific Decadal
Oscillation (PDO) between 1977 and the mid-1990s (Kayano et al., 2009; Jacques-Coper and Garreaud, 2015; González-Reyes
et al., 2017). Additionally, the 2010 Aconcagua streamflow breakpoint is likely driven by the onset of the megadrought, which
also affected the 2004 change points in the Limarí and Choapa Basins where lower precipitation was observed even before the
megadrought.

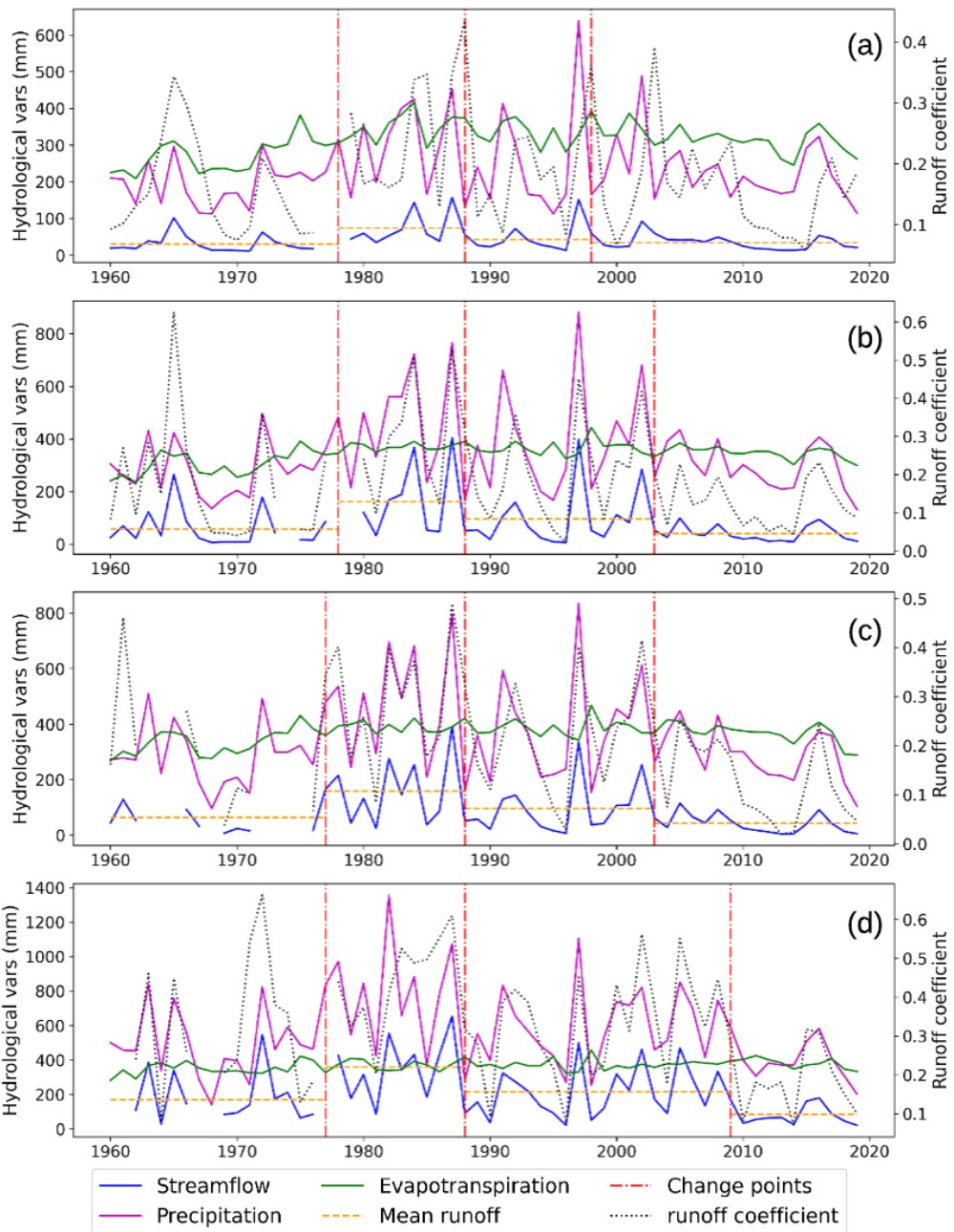

**Figure 4. Annual streamflow, precipitation, evapotranspiration, and runoff coefficient during the complete period (1960-2020) for Elqui (a), Limarí (b), Choapa (c), and Aconcagua (d) Basins, respectively. The vertical red line indicates the years where significant change points (P value < 0.05) on streamflow distribution are detected by the Buishand test.**

Regarding water use, breakpoints were observed in Elqui and Limarí in 1988 and 1992, respectively, mainly associated to the growth of the agricultural sector (Fig. 5a and b). In the Aconcagua basin, a breakpoint occurred in 1985 due to intensified water use by the mining and agriculture sectors (Fig. 5d). Meanwhile, in the Choapa basin, a significant increase in mining water consumption since 2000 explains the time series breakpoint observed in that year (Fig. 5c). The 1998 Elqui Basin streamflow breakpoint may be attributed to the construction of a dam upstream from the gauge station considered in this study (Fig. 5a). Based on these results, we used the 1988 streamflow breakpoint detected in all basins to define the low-influence period of 1960-1988. In consequence, the evaluation period was defined between 1988 and 2020, characterised by greater anthropic intervention and by the megadrought in its second half.

By comparing the hydroclimatic conditions of the study basins during the low-influence and evaluation periods, we see that the mean annual precipitation declined between 0.04 to 15.89% during these periods (Table 1). In contrast, the mean annual streamflow decreased by a range of 13.97 to 37.25%. If we examine summer streamflow, when agricultural water consumption is more intense, a reduction of 24.25 to 46.1% is observed. While the Aconcagua basin features the largest decrease in precipitation, the Choapa basin has the largest decrease in streamflow.

| Basin | Mean annual precipitation (mm) | | | Mean annual streamflow(mm) | | | Mean summer streamflow(mm) | | |
|---|---|---|---|---|---|---|---|---|---|
| | Low-influence period | Evaluation period | Difference | Low-influence period | Evaluation period | Difference % | Low-influence period | Evaluation period | Difference % |
| Elqui | 232.83 | 232.73 | -0.04% | 45.53 | 39.17 | -13.97% | 28.66 | 21.71 | -24.25% |
| Limarí | 355.13 | 336.78 | -5.17% | 95.91 | 66.92 | -30.23% | 54.5 | 33.87 | -37.85% |
| Choapa | 371.16 | 327.76 | -11.69% | 106.41 | 66.77 | -37.25% | 68.09 | 36.7 | -46.10% |
| Aconcagua | 634.61 | 533.76 | -15.89% | 258.42 | 173.87 | -32.72% | 193.29 | 119.82 | -38.01% |

**Table 1: Average annual precipitation, average annual streamflow, and average summer season streamflow for each basin in the low-**
**influence reference period (1960-1988) and the evaluation period (1988-2020).**

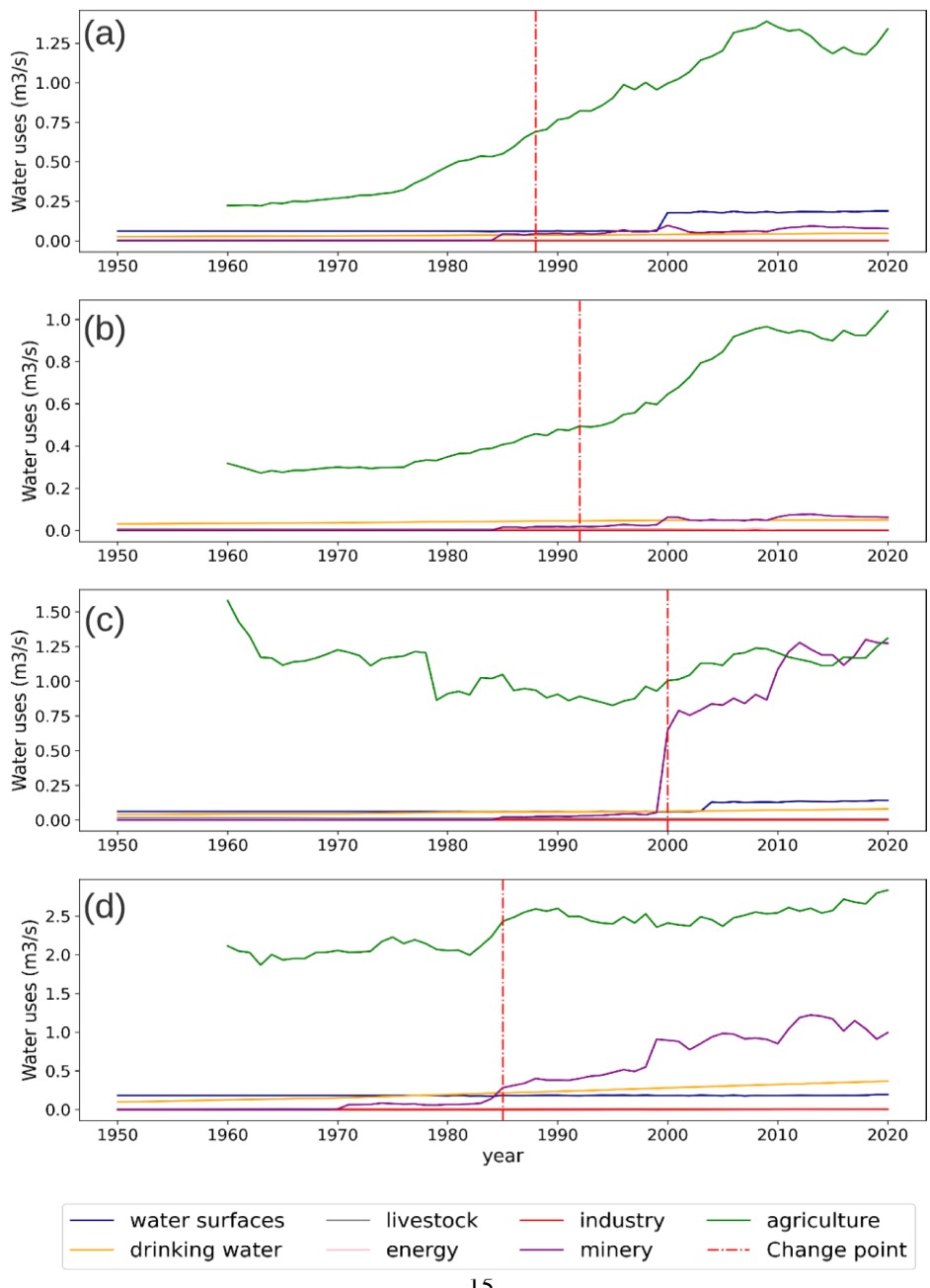

254                                            15

 **Figure 5. Time series of water uses from different human activities in Elqui (a), Limarí (b), Choapa (c), and Aconcagua (d) basins,**
**respectively. These time series include water uses for industrial, agriculture, mining, energy, animals, water surfaces, and drinking**
**water sectors. The red line indicates a breakpoint in the total water use distribution.**
**3.2 Near-natural streamflow estimation**
Near-natural simulated streamflow during the low-influence and evaluation periods for each basin is presented in Fig. 6. The
selected models (Sect. 2.3) were based on streamflow values without the Box-Cox transformation, since the transformed data
led to a reduction in model performance across all basins (Appendix A). The summer season estimations obtained from Eq. 2
had good performances during the training period, with mean biases of 0 to 5% and $r^2$ ranging from 0.81 to 0.89 for the different
basins. The winter season models resulted in lower performance, with mean biases of 0 to 0.63% and $r^2$ ranging from 0.61 and
0.93 among the study basins.

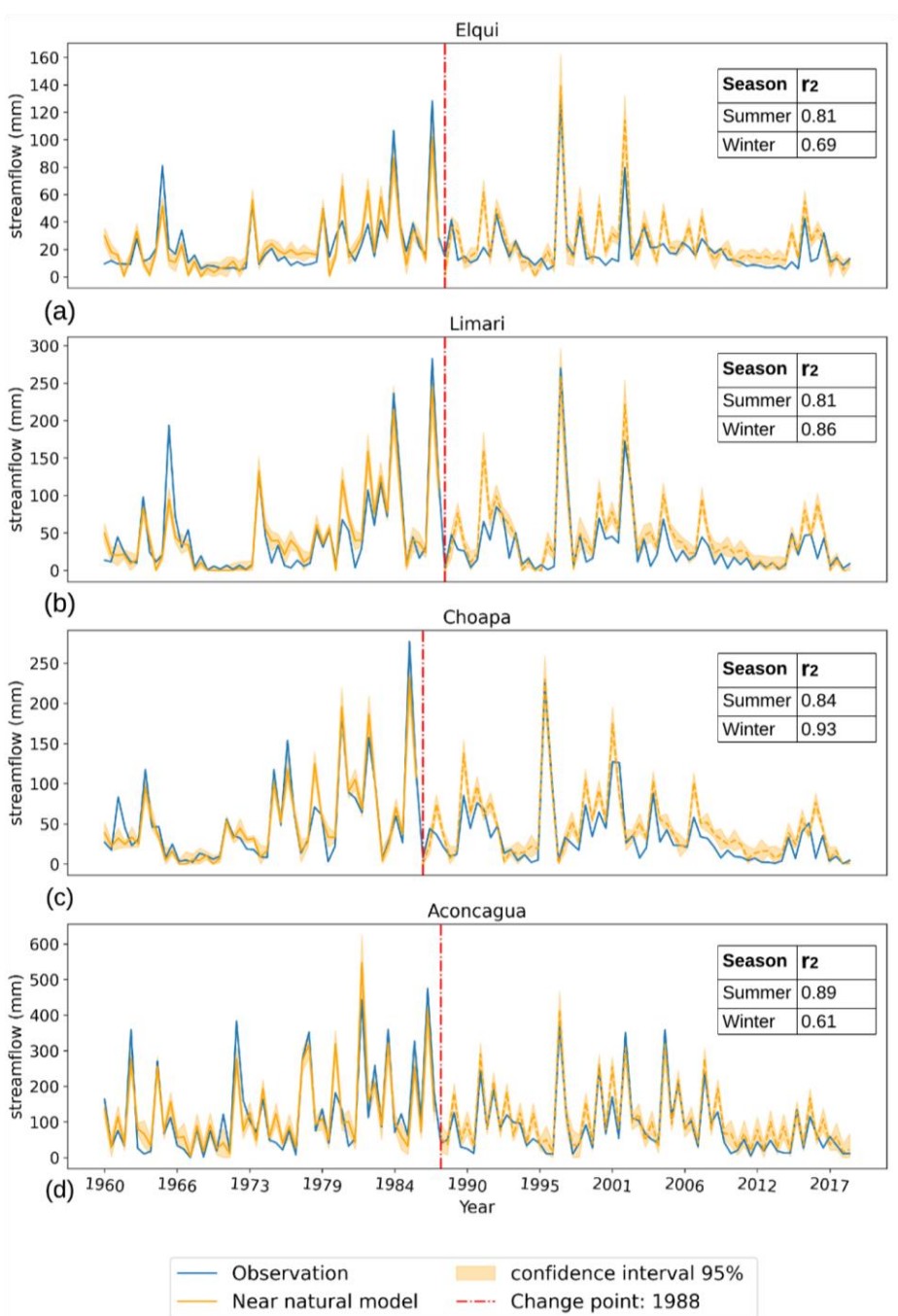

 **Figure 6. The observed (continuous blue line) and near-natural simulated seasonal streamflow (continuous and dashed yellow line) for**
**Elqui (a), Limarí (b), Choapa (c), and Aconcagua (d) basins, respectively. The continuous yellow line represents the simulated**
**streamflow during the reference period, whose $r^2$ is presented on the legend. The dashed yellow line is the simulated streamflow during**
**the evaluation period (defined by the change point in 1988). The yellow ban represents the 95% confidence interval of the simulated**
**streamflow.**
To examine the potential influence of non-stationary catchment response during the megadrought on the interpretation of our
results, Table 2 shows the rainfall-runoff ratios during the evaluation period before (1988-2010) and after the megadrought onset
(2010-2020). These results indicate that the mean rainfall-runoff ratios declined across the upper and lower sections (defined by
up-stream and attribution stations from Fig. 1a, respectively) of all basins during the megadrought, however, the reduction in the
upper sections (with low human intervention), mostly attributed to endogenous runoff mechanisms and hydrological memory,
is less significant than those observed downstream (intervened basin). The changes in downstream rainfall-runoff ratios are
nearly four times greater than the upper stream changes in the Aconcagua and Elqui basins, more than twice in Choapa, and 1.6
times greater in the Limarí basin. This indicates that while endogenous runoff mechanisms, such as hydrological memory, may
contribute to larger streamflow deficits during prolonged drought in near-natural basins, human activities in the downstream
basins are inducing a larger impact on runoff generation during the megadrought.

| | Basin | Elqui | | Limarí | | Choapa | | Aconcagua | |
|---|---|---|---|---|---|---|---|---|---|
| | Section | Upper | Lower | Upper | Lower | Upper | Lower | Upper | Lower |
| **Period** | 1988 -2010 | 0.42 | 0.19 | 0.41 | 0.18 | 0.58 | 0.21 | 0.75 | 0.33 |
| | 2010 -2020 | 0.38 | 0.12 | 0.31 | 0.11 | 0.43 | 0.09 | 0.66 | 0.18 |
| | Difference | 9.03% | 34.3% | 25.2% | 40.4% | 24.94% | 58.33% | 11.94% | 46.21% |

**Table 2: Average annual runoff coefficient during the evaluation period before the megadrought onset (1988-2010) and during the**
**megadrought (2010-2020) for the upper and lower sections of each basin. The difference between the two periods relative to 1988-2010**
**is shown in the third row.**
**3.3 The impacts of climate and human activities on streamflow**
During the complete evaluation period, the near-natural simulated streamflow is higher than the observed streamflow in all the
cases (Fig. 6) with mean differences ranging from 39.4% in the Limarí basin (near-natural annual runoff of 93.23 mm and

observed annual runoff of 66.91 mm) to 20.7% in the Aconcagua basin (near-natural annual streamflow of 210 mm and observed annual runoff of 173.86 mm).

The relative impacts of climate and human activities on summer streamflow reductions during the evaluation period is presented in Fig. 7. This figure shows the annual anomalies of precipitation, observed and near-natural simulated summer streamflow, as well as the human-induced streamflow reduction obtained as the difference of the latter two (Eq. 4). The results for the annual fluxes are presented in Appendix B.

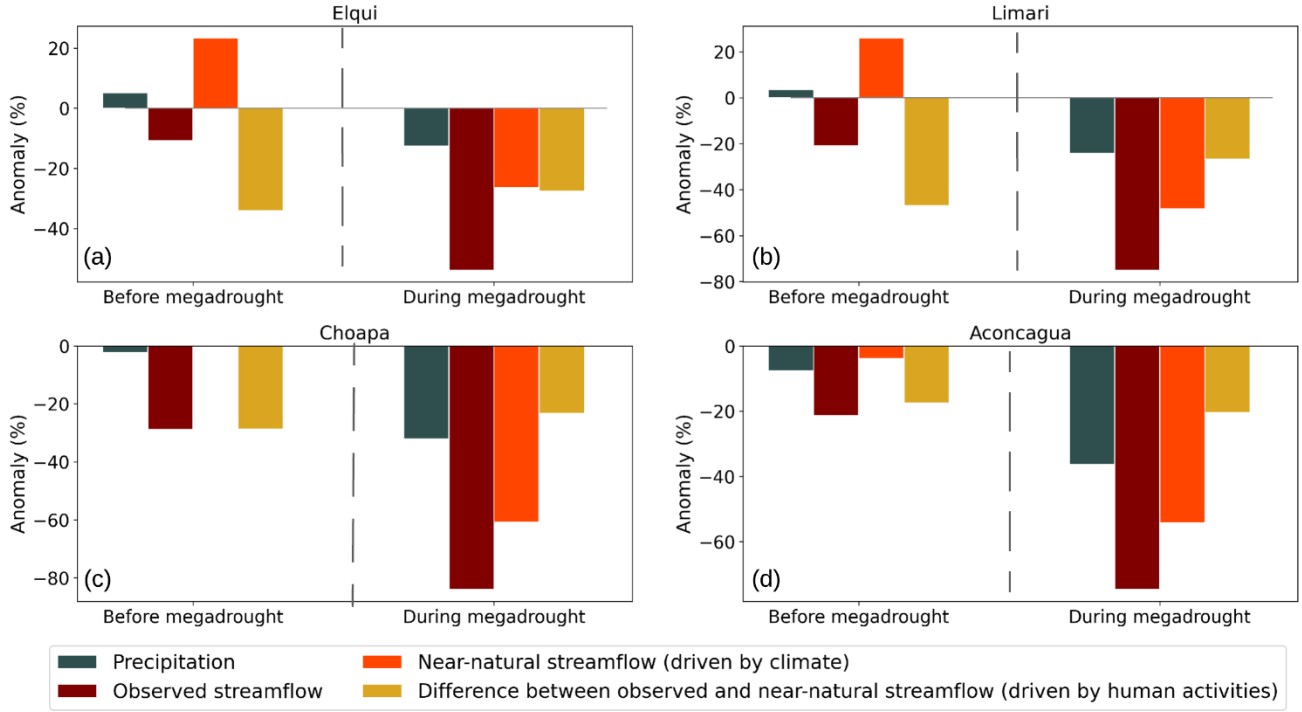

**Figure 7: Anomalies in annual precipitation, observed and near-natural summer streamflow, and the difference between the latter two, which represent the human-induced streamflow anomaly for Elqui (a), Limarí (b), Choapa (c), and Aconcagua (d) basins. The anomalies are presented for the evaluation period before and after the megadrought onset (1988-2009 and 2010-2020, respectively).**

**For each flux, the anomalies are computed as the percentage difference with respect to their mean values during the reference period (1960-1988).**

Before the megadrought onset, annual precipitation varied between 5 to -7.6% with respect to the reference period among the study basins. The near-natural summer streamflow during that period followed the direction of the annual precipitation anomalies, with anomalies between 23 to -4% across basins. During that period, the observed summer streamflow -accounting for full climatic and human influence- decreased by 10-28%. This indicates that water uses for human activities were the main driver factor of summer streamflow reduction before the megadrought onset, causing up to 100% reduction in Elqui, Limarí and Choapa, and 82% in the Aconcagua Basin, respectively.

After the megadrought onset, the relative impact of precipitation deficits and human activities on streamflow depletion changed. The annual precipitation anomalies during the megadrought varied between -13 to -36% across basins, while the near-natural streamflow estimates presented anomalies between -26% to - 61%. During this period, the observed summer streamflow featured anomalies of -54% to -84%. This indicates that precipitation deficits dominate the streamflow reductions, however, there is still a relevant reduction attributed to human activities, representing 51%, 29%, 27%, and 27% of the total summer streamflow reduction in Elqui, Limarí, Choapa, and Aconcagua Basin, respectively.

Particularly noteworthy is the Aconcagua basin case, where, in absolute terms, the human induced summer streamflow reduction during the megadrought (corresponding to an absolute value of 39.47 mm) was higher than during the period before the megadrought (33.78 mm). This has happened despite the significantly lower water availability during the megadrought, where near-natural summer streamflow was 88.61 mm, which corresponds to less than half of the near-natural summer flow before the megadrought onset (185,73 mm). This apparent contradiction may be attributed to the Aconcagua's increased total water consumption during the megadrought, led by intensified agricultural water demand (Fig. 5a).

Consistently with the summer seasons, near-natural annual streamflow before the megadrought followed precipitation patterns, with anomalies between 22 to -5% across basins (Fig. A1). During that period, the observed annual streamflow varied between -2 to -20% across basins. Water uses for human activities were the driver factor of streamflow reduction before the megadrought onset, causing up to 100% of reduction in Elqui, Limarí and Choapa, and 71% in the Aconcagua Basin, respectively. After the megadrought onset, the observed streamflow featured anomalies of -47 to -71%. From these streamflow deficits, a 44% to 75 %

of the reduction is attributed to climatic-factors (i.e., anomalies represented by the near-natural simulated streamflow), while the
remaining 25 to 56% is attributed to human activities.

### 3.4 The impacts of human activities on hydrological drought events

The selected hydrological drought events for each basin are shown in Fig. 8. By contrasting the observed and near-natural time
series, the climate-induced and human-induced droughts are distinguished. The meteorological megadrought (2010-2020) is
associated to several hydrological drought events, as evidenced by the observed streamflow time series. However, the
megadrought does not seem to have such a persistent and intense effect on the near-natural streamflow.
The human impact on hydrological droughts (computed as the difference between observed and near-natural streamflow drought
events) is evident in the duration and intensity of drought events (Table 3). Elqui, Limari, Choapa, and Aconcagua have 10, 13,
13 and 7 extra seasons in drought duration, respectively, and close to double of streamflow deficits. In general, more drought
events (Limari, Choapa and Aconcagua) with a larger average time duration (Elqui and Choapa) and average deficit (Elqui,
Choapa and Aconcagua) have occurred in the observed scenario compared to the near-natural scenario. The largest drought event
in each basin occurred during the megadrought. Across all basins, the human activities led to an increase in the maximum
duration of hydrological droughts, with maximum values ranging between 10 to 12 seasons, in contrast to 4 to 6 seasons
experienced in the near-natural cases. In particular, this translates to 5 or 6 years of continuous streamflow below the Q70
threshold during the megadrought. The human influence over hydrological drought varies between the different drought
characteristics, but in most cases it causes drought intensification, leading to an increase of 25.93 to 44.83% of the total drought
events and an increase of 17.89 to 61.66% of the total streamflow deficit. The negative percentage difference in mean duration
or mean deficit reported for Limari and Aconcagua basins is due to a greater number of shorter events. However, considering
that the total number of events is larger in the observed scenario, this is not indicative of an alleviation of the drought.
When analyzing drought characteristics separately before and during the megadrought (Appendix C), Elqui exhibits a low human
impact before the megadrought onset and it notably increases during the megadrought, contributing to 57.14% of total drought
events and 70.82% of the observed deficit. In contrast, Limari, Choapa, and Aconcagua show a more stable human contribution
to drought characteristics before and during the megadrought, with a decrease of human contribution to total events close to 25%
(all basins), a decrease in human contribution to total deficit (Limarí basin) and a slight increase contribution to total deficit
during the megadrought (Choapa basin).

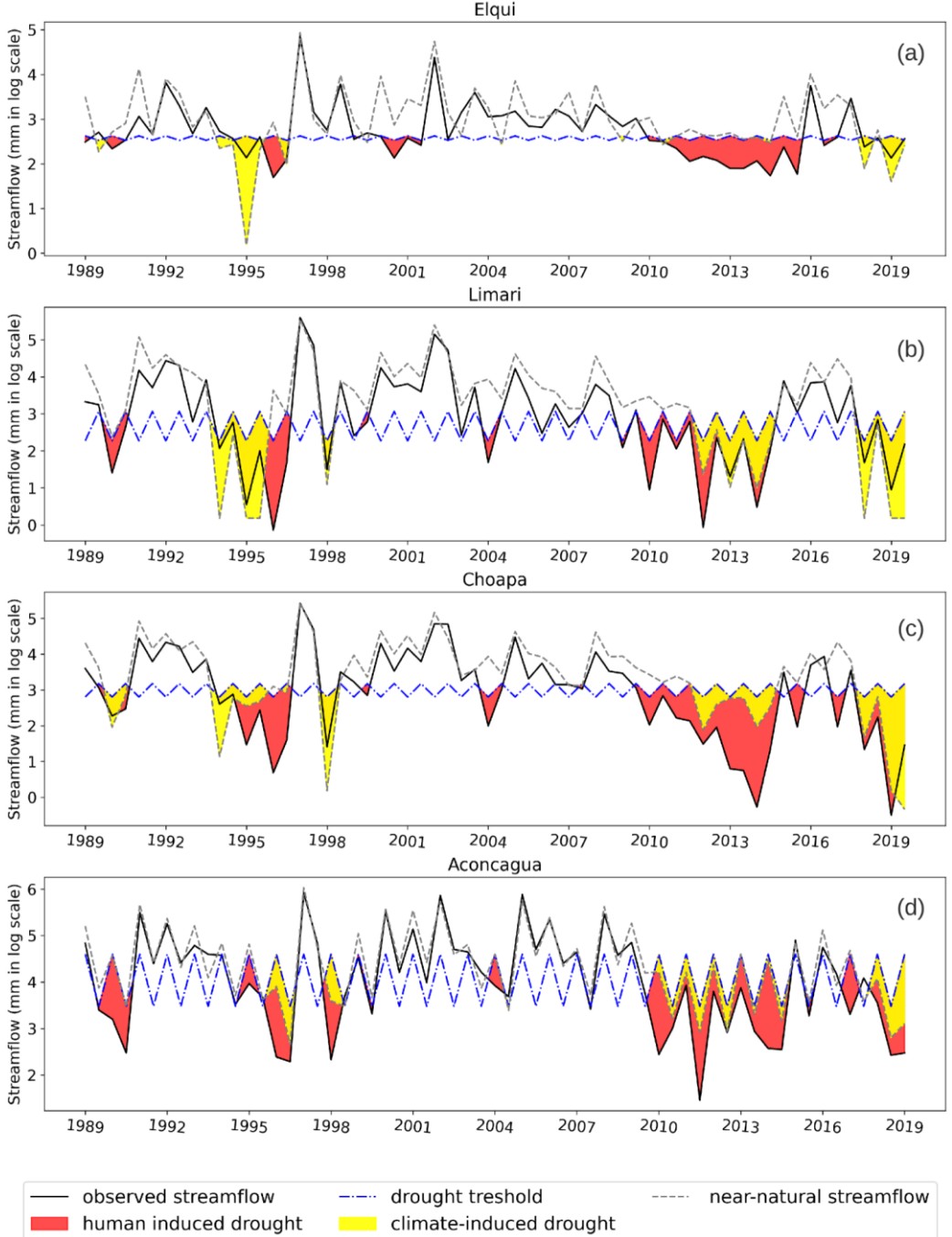


Figure 8. Observed and near-natural streamflow and hydrological drought events during the evaluation period (1988–2020) for Elqui (a), Limarí (b), Choapa (c), and Aconcagua (d) basins, respectively.

| Basin | Hydrological Drought | Frequency | Duration (seasons) | | | Deficit (mm) | | |
|---|---|---|---|---|---|---|---|---|
| | | | Total seasons | Max duration | Average duration | Total deficit | Max deficit | Avergae deficit |
| Elqui | Near- natural | 10.00 | 16.00 | 4.00 | 1.60 | 51.61 | 20.55 | 5.16 |
| | Observed | 10.00 | 26.00 | 12.00 | 2.60 | 95.30 | 54.77 | 9.53 |
| | diff % | 0.00% | 38.46% | 66.67% | 38.46% | 45.84% | 62.47% | 45.84% |
| Limari | Near- natural | 5.00 | 16.00 | 6.00 | 3.20 | 157.34 | 51.69 | 31.47 |
| | Observed | 10.00 | 29.00 | 10.00 | 2.90 | 191.62 | 77.05 | 19.16 |
| | diff % | 50.00% | 44.83% | 40.00% | -10.34% | 17.89% | 32.92% | -64.22% |
| Choapa | Near- natural | 7.00 | 19.00 | 6.00 | 2.71 | 181.43 | 58.31 | 25.92 |
| | Observed | 11.00 | 32.00 | 11.00 | 2.91 | 355.70 | 142.27 | 32.34 |
| | diff % | 36.36% | 40.63% | 45.45% | 6.70% | 48.99% | 59.01% | 19.85% |
| Aconcagua | Near- natural | 7.00 | 20.00 | 6.00 | 2.86 | 411.97 | 133.27 | 58.85 |
| | Observed | 12.00 | 27.00 | 10.00 | 2.25 | 1074.65 | 415.00 | 89.55 |
| | diff % | 41.67% | 25.93% | 40.00% | -26.98% | 61.66% | 67.89% | 34.28% |

Table 3: drought characteristics for each basin considering the observed and simulated near natural streamflow during the evaluation period (1988-2020). The third row for each basin represents the human influence on drought characteristics as the percentage difference between the observed and the naturalized scenario

## 4 Discussion

### 4.1 Impact of increased human activities on water availability

During the megadrought, precipitation deficits have played a more significant role on the decrease in annual streamflow than anthropic factors, however, human activities still account for approximately 27 to 29% of the streamflow reduction in the Aconcagua, Choapa, and Limarí basins and 51% in Elqui, the basin least affected by the meteorological megadrought.

Human activities have intensified since the 1980s, driven by rising water demand from economic activities, population growth, and land use changes (Fig. 5a), despite the precipitation deficits and streamflow reduction during the megadrought. In general, the basins with the greatest increases in total water consumption during the evaluation period also exhibit higher human influence

in the reduction of streamflow. Elqui and Limari exhibited the most significant relative increase in total water consumption,
primarily driven by a substantial rise in agriculture consumption from 1989 to 2010, while Choapa almost duplicated its total
water consumption during the 2000-2010 decade due to mining operations. It is noteworthy that agriculture and mining water
consumption continued to rise during the megadrought.
This suggests that total water consumption from surface and groundwater sources has been somehow inelastic to the surface
water deficits. In the Aconcagua basin, the human-induced streamflow reduction expressed as mm increased during the
megadrought, while in the other three basins was slightly smaller compared to the period prior to the megadrought (Fig. 7). This
finding can be explained by an initial reduction in agricultural water consumption during the first years of the megadrought,
which was later reversed (Fig. 5a) by higher extractions of groundwater sources in the subsequent years (Taucare et al., 2020;
Duran-Llacer et al., 2020).
Groundwater sources play a crucial role in streamflow within this study region, and the declines of groundwater levels caused
by meteorological droughts and water extractions have critical impacts on water accessibility in rural areas (Crespo et al., 2020;
Taucare et al., 2020; Alvarez-Garreton et al., 2021; 2023; 2024). These declines can also lead to the disconnection between
surface and underground water sources, leading to a decrease in soil moisture conditions (agricultural drought) and the
desiccation of rivers and lakes (Duran-Llacer et al., 2022; Muñoz et al., 2020). This exacerbates hydrological drought, delaying
the recovery of catchments from drought episodes. Also, irrigation water extraction shifts from surface to groundwater sources,
intended to alleviate megadrought impacts, also promotes the inelastic behaviour of water consumption rates. In fact, new surface
and underground water use rights have been granted during the megadrought (Barría et al., 2021b). This has led to increases in
water stress levels and reduction of groundwater reservoirs, which could ultimately lead to an absolute day zero (Alvarez-
Garreton et al., 2024).
Despite a general decrease in the impact of human influence on streamflow reductions between the pre-megadrought and
megadrought periods, the Limari, Choapa, and Aconcagua basins show a relatively stable human contribution to drought
characteristics before and during the megadrought, while the Elqui basin experiences a notable increase in human contribution.
These observations highlight two key insights. First, they suggest that human activities have a greater influence on drought
conditions and characteristics than the solely relative impact of human activities on total streamflow reductions. In the context
of meteorological drought, increased and inelastic human water demand exacerbates streamflow reductions, causing them to
exceed hydrological drought thresholds in terms of both magnitude and frequency. Second, the increase in human contribution
to drought characteristics in the Elqui basin leads it to similar patterns of hydrological drought conditions than the other basins,
despite having lower precipitation deficits. This suggests that the role of human water demands is particularly relevant in semi-
arid basins with limited precipitation and high interannual variability in terms of precipitation regime, such as Elqui.
Consequently, highly intervened basins in semi-arid regions are more susceptible to experiencing severe hydrological droughts
during periods of precipitation deficits. These findings align with the observations of Huang (2016), who highlighted that
sustainable agricultural development is threatened in arid and semi-arid regions due to limited available water resources, and
with Saft et al. (2016), who demonstrated that aridity is a crucial factor influencing streamflow sensitivity to interdecadal climate
variability.

**4.2 Drought vulnerability**

Hydrological drought vulnerability is associated with those conditions that cause an increase in the frequency, duration, and
intensity of the hydrological droughts when a precipitation deficit threat is faced. Vulnerability should be addressed by looking
for sensitivity variables that come from the biophysical basin's characteristics, such as aridity, location, geomorphology,
hydrological regime, natural land cover, and snow and glacier cover (Saft et al., 2015; Van Loon and Laaha, 2015), and human
activities such as management and extraction of water, land use, land cover changes, urbanisation, between others (Barría et al.,
2021a; Van Loon et al., 2016, 2022).
As discussed in Sect. 4.1, human activities have intensified streamflow deficits during the megadrought. Human activities that
affect catchment vulnerability in central Chile include groundwater extractions (Taucare et al., 2020), overallocation of water
use rights (Alvarez-Garreton et al., 2021; Barría et al., 2021a), and continuous land use change for agricultural purposes
(Madariaga et al., 2021). For example, agriculture is sometimes established on hillsides with high slopes, exacerbating water
consumption problems and changing runoff mechanisms. In the entire Aconcagua basin, the water consumption of avocado
plantations has increased 15% between 2014 and 2020, reaching almost 4.8 $m^3$/s, while citrus plantations have increased 67-
70% in the Elqui and Limarí basins since 2010, reaching 1.8 $m^3$/s of water consumption in the Limarí basin. This reveals that
irrigated agriculture has been inelastic to the precipitation deficits during the megadrought. Human activities in these basins are
adapting to less water availability in ways that are leading to aggravated water scarcity problems, which is considered in the
literature as maladaptation (Schipper, 2020).

Precipitation deficits and human activities including human-induced maladaptation processes have broad, complex and exacerbated impacts on society and ecosystems. For example, agricultural practices may worsen water scarcity problems and contribute to soil erosion and sediment transport (Owens, 2020), further degrading ecosystem health. The intensified streamflow deficits have disrupted watercourses and contribute to tree mortality (Miranda et al., 2020). Additionally, thousands of people have lost access to domestic water services (Muñoz et al., 2020), leading to a large spending on water cistern trucks (Alvarez-Garreton et al., 2023). These impacts reveal that there is still a gap in understanding how human activities contribute to catchment vulnerability to hydrological droughts and how their influence on the hydrological cycle can be effectively included in drought management (Anne F. Van Loon et al., 2016). In the case of Chile, previous studies have shown that the current water management policy inadequately addresses the physical constraints of surface and groundwater availability, contributing to an inadequate prevention of water stress conditions (Alvarez-Garreton et al., 2023). This calls for urgent modifications in the water management system to ensure sustainable water use and prevent the exacerbation of water stress conditions in the region.

**4.3 Study limitations**

Our approach and insights are based on attribution exercises that compare the observed streamflow and a naturalised simulation of it, which permits to isolate the effect of human activities. In this study, the near-natural simulation was done by using regression statistical methods, which have limitations that should be considered: they do not explicitly account for the physical mechanisms of runoff generation, they rely solely on precipitation as a predictor and they consider a linear relationship between the variables. Although the attribution exercise is still consistent, this methodological limitation prevents to drawing conclusions regarding the physical mechanisms involved in streamflow reduction during droughts. To enable a physical interpretation, and likely a better representation of streamflow generation and memory effects, future studies should advance into implementing physically-based models to perform the attribution exercises.

Independently of the adopted model, the streamflow estimations have uncertainties that can mask some of the human influence effects in the attribution exercise. In order to visualize this potential artefact, Fig. 6 shows the streamflow estimations with a 95% confidence interval. These plots, in general, show that the lowest values of naturalized streamflow are above the observed time series. Anyway, considering the lower performance of the winter models in some catchments and that the summer season concentrates most human intervention due to agricultural activities, we have primarily focused on exploring the results of this season (Fig. 7).

Considering the evidence of potential climate-driven non-stationarities on streamflow generation during the megadrought in Chilean catchments (Alvarez-Garreton et al., 2021), the attribution of human activities as the driving factor of the intensified streamflow reduction should then interpreted carefully. The intensification in streamflow reduction is attributed to the combination of human activities, natural hydrological processes, and the potential effects of non-stationarity catchment response. Since the upper catchment sections have a lower human influence (but still influenced) than the downstream sections, the larger streamflow decrease during the megadrought (compared to the previous period) in these sections may be mostly (but not fully) attributed to non-stationarity in basin response during protracted droughts (consistent with Saft et al., 2015; Alvarez-Garreton et al., 2021). However, the downstream sections feature an even larger streamflow reduction during the megadrought compared to the reduction in the upper sections (Table 2). This is consistent to the added effect of human activities on streamflow reduction, which have maintained water consumption despite the reduced water availability (Fig. 5).

## 5 Conclusions

The megadrought in central Chile has been the longest dry period over the last centuries. The study basins featured a range of 16 to 41% in mean annual precipitation deficits during this period, whereas the deficits in observed streamflow were significantly larger. The Elqui, Limarí, Choapa, and Aconcagua basin experienced deficits in summer streamflow of 54%, 75%, 84%, and 75%, respectively.

Our findings indicate that human activities were the main driving factor of streamflow reduction before the megadrought onset. During the megadrought, human activities still accounted for a significant portion of streamflow reduction, ranging from 27 to 51%. The impact of human activities on hydrological drought characteristics was substantial, leading to more than double the recurrence, duration, and intensity of droughts in some basins.

Human activities in these basins have shown limited adaptation to the decrease in water availability. The increase in human water demand, often inelastic to the decreased surface water availability, makes basins more vulnerable to severe hydrological droughts when precipitation deficits are faced, especially on semi-arid basins with water availability constraints.

This paper demonstrates that during long and persistent dry periods, human activities in basins in central Chile have intensified drought propagation, by increasing both the intensity and the duration of hydrological droughts. This highlights the importance of understanding the impacts of human activities on drought propagation, and to consider such evidence in water management

policies. In particular, to prevent implementing maladaptive measures, the feedback loop between water usage, human activities,
and the hydrological system should be considered in the adaptation strategies. These considerations are particularly important
not only in Chile but also in other regions worldwide, where the dry signal is consistent and expected to persist.

**Data availability**

The CR2MET dataset were obtained from the Center for Climate and Resilience Research website at https://www.cr2.cl/datos-
productos-grillados (last access: 20 September 2023). The water use data was obtained upon request from the Center for Climate
and Resilience Research website at https://seguridadhidrica.cr2.cl (last access: 20 September 2023). The streamflow data were
obtained from CAMELS-CL dataset (Alvarez-Garreton et al., 2018), available at the Center for Climate and Resilience Research
website at https://camels.cr2.cl (last access: 20 September 2023).

**Author contributions**

NA, CAG and AM conceived the idea of the research. NA performed the analyses. NA and CAG wrote much of the manuscript.
All the authors reviewed early manuscript drafts and the final draft.

**Competing interests**

The contact author has declared that none of the authors has any competing interests.

**Acknowledgements:**

This research has been developed within the framework of the Center for Climate and Resilience Research (CR2,
ANID/FONDAP/1522A0001), the research project ANID/FSEQ210001, ANID/FONDECYT/1201714 and
ANID/FONDECYT/11240924.

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

**Appendix A.**
This appendix presents the outcomes of a comprehensive evaluation of various regression models, considering the seasonal
runoff as a dependent variable. The objective was to identify the key climate factors influencing the streamflow response in
the studied basins. Variables such as precipitation in different seasons, evapotranspiration, temperature, and the interaction
between temperature and precipitation were used. Additionally, a model incorporating a Box-Cox transformation of the
dependent variable (runoff) was examined to achieve a normal distribution in the variable.
After rigorous testing, it is noteworthy that the majority of the models demonstrated a singular dependency on precipitation
(P). We chose the model with a higher r2, and all variables were statistically significant at a p-value of 0.05 In Summer (Table
A1) this condition is achieved with model 1 (eq 2 of sect 2.3.2), where the summer runoff is modelled based on the winter
precipitations. In winter (table A2) the condition is achieved in model 2 (eq 3 of sect 2.3.2) where the runoff depends on the
winter precipitation of the present year (t) and the annual precipitation of the previous year (t-1).

| Variables/ Models | Dependent variable: Summer runoff | | | | | | | | | | | | | | | | | | | |
| --- | --- | --- | --- | --- | --- | --- | --- | --- | --- | --- | --- | --- | --- | --- | --- | --- | --- | --- | --- | --- |
| | Model 1 | | | | Model 2 (runoff Box-Cox) | | | | Model 3 | | | | Model 4 | | | | Model 5 | | | |
| | Elqui | Lim | Choap | Acon | Elqui | Lim | Choap | Acon | Elqui | Lim | Choap | Acon | Elqui | Lim | Choap | Acon | Elqui | Lim | Choap | Acon |
| const | -19.80*** | -70.05*** | -56.80*** | -61.87*** | 1.52*** | 0.12 | 1.38*** | 3.57*** | -8.89 | -55.36** | -59.92*** | -111.17*** | 8.59 | -3.46 | -3.18 | 39.82 | 86.77 | 174.87 | -158.72 | -218.17 |
| | (-5.49) | (-12.5) | -11.34 | (-28.38) | (-0.17) | (-0.4) | (-0.36) | (-0.33) | (-10.11) | (-19.89) | (-18.7) | (-39.06) | (-22.99) | (-57.02) | (-59.14) | (-110.76) | (-90.18) | (-211.48) | (-186.45) | (-277.96) |
| P_winter(t) | 0.31*** | 0.45*** | 0.39*** | 0.49*** | 0.01*** | 0.01*** | 0.01*** | 0.01*** | 0.31*** | 0.46*** | 0.39*** | 0.47*** | 0.33*** | 0.46*** | 0.40*** | 0.49*** | 0.31*** | 0.47*** | 0.39*** | 0.47*** |
| | (-0.03) | (-0.04) | (-0.03) | (-0.05) | 0 | 0 | 0 | 0 | (-0.03) | (-0.04) | (-0.03) | (-0.05) | (-0.03) | (-0.04) | (-0.03) | (-0.05) | (-0.03) | (-0.04) | (-0.03) | (-0.05) |
| P_summer(t) | | | | | | | | | -0.14 | -0.24 | 0.07 | 0.56* | | | | | | | | |
| | | | | | | | | | (-0.11) | (-0.25) | (-0.32) | (-0.32) | | | | | | | | |
| ET_summer | | | | | | | | | | | | | -0.2 | -0.41 | -0.28 | -0.57 | | | | |
| | | | | | | | | | | | | | (-0.16) | (-0.34) | (-0.3) | (-0.6) | | | | |
| T_mean_summer | | | | | | | | | | | | | | | | | -9.9 | -19.78 | 8.55 | 12 |
| | | | | | | | | | | | | | | | | | (-9.29) | (-18.13) | (-16.16) | (-30.65) |
| TxP | | | | | | | | | | | | | | | | | -0.01 | -0.02 | 0 | -0.06 |
| | | | | | | | | | | | | | | | | | (-0.01) | (-0.02) | (-0.03) | (-0.04) |
| | | | | | | | | | | | | | | | | | | | | |
| Observations | 27 | 27 | 22 | 26 | 27 | 27 | 22 | 26 | 27 | 27 | 22 | 26 | 27 | 27 | 22 | 26 | 27 | 27 | 22 | 26 |
| R2 | 0.81 | 0.84 | 0.89 | 0.81 | 0.82 | 0.74 | 0.74 | 0.61 | 0.82 | 0.84 | 0.89 | 0.83 | 0.82 | 0.85 | 0.89 | 0.81 | 0.83 | 0.85 | 0.89 | 0.83 |
| Adjusted R2 | 0.8 | 0.83 | 0.88 | 0.8 | 0.81 | 0.73 | 0.73 | 0.59 | 0.8 | 0.83 | 0.88 | 0.81 | 0.8 | 0.84 | 0.88 | 0.8 | 0.81 | 0.83 | 0.87 | 0.81 |
| Residual Std. Error | 14.2 | 31.74 | 25.29 | 62.87 | 0.45 | 1.01 | 0.8 | 0.74 | 14.02 | 31.8 | 25.91 | 60.29 | 14.03 | 31.47 | 25.38 | 63 | 13.92 | 31.58 | 26.4 | 61.35 |
| F Statistic | 103.34*** | 130.43*** | 156.66*** | 99.65*** | 110.49*** | 69.99*** | 57.83*** | 36.78*** | 53.80*** | 65.42*** | 74.62*** | 55.72*** | 53.75*** | 67.06*** | 78.19*** | 50.07*** | 37.05*** | 23.73*** | 18.66*** | 13.94*** |
| Note: | *p<0.1; **p<0.05; ***p<0.01; std_dv in () | | | | | | | | | | | | | | | | | | | |


**Table A1: results of multiple regression equations tested for representing near-natural streamflow during the reference period in the**
**summer season**

| Variables/ Models | Model 1 | | | | Model 2 | | | | Model 3 runoff (Box-Cox) | | | | Model 4 | | | | Model 5 | | | |
|---|---|---|---|---|---|---|---|---|---|---|---|---|---|---|---|---|---|---|---|---|
| | Elqui | Lim | Choap | Acon | Elqui | Lim | Choap | Acon | Elqui | Lim | Choap | Acon | Elqui | Lim | Choap | Acon | Elqui | Lim | Choap | Acon |
| **const** | 11.90*** | -15.45** | -10.53** | -11.17 | -5.22 | -30.94*** | -20.01*** | -31.44 | 1.41*** | 0.75 | 1.13** | 0.16 | -9.53** | -35.18*** | -7.67 | -16.03 | -32.84 | -117.70* | -45.62 | -116.79 |
| | (-3.67) | (-6.23) | (-4.02) | (-14.14) | (-3.49) | (-6.85) | (-4.57) | (-20.08) | (-0.42) | (-0.44) | (-0.52) | (-1.31) | (-4.31) | (-12.07) | (-7.95) | (-53.56) | (-20.13) | (-58.61) | (-40.73) | (-82.56) |
| **P_winter(t)** | 0.04* | 0.19*** | 0.16*** | 0.14*** | 0.02 | 0.17*** | 0.16*** | 0.14*** | 0.01*** | 0.01*** | 0.01*** | 0.01*** | -0.01 | 0.17*** | 0.17*** | 0.14*** | 0.16 | 0.57*** | 0.23* | 0.27* |
| | (-0.02) | (-0.02) | (-0.01) | (-0.02) | (-0.01) | (-0.02) | (-0.01) | (-0.02) | 0 | 0 | 0 | 0 | (-0.02) | (-0.02) | (-0.01) | (-0.03) | (-0.11) | (-0.17) | (-0.11) | (-0.13) |
| **P_winter(t-1)** | | | | | 0.09*** | 0.06*** | 0.03** | 0.03 | 0.01*** | 0.01*** | 0.00*** | 0.00*** | 0.07*** | 0.05* | 0.05*** | 0.04 | 0.08*** | 0.05*** | 0.03** | 0.03 |
| | | | | | (-0.01) | (-0.02) | (-0.01) | (-0.02) | 0 | 0 | 0 | 0 | (-0.02) | (-0.03) | (-0.01) | (-0.03) | (-0.02) | (-0.02) | (-0.01) | (-0.03) |
| **ET_winter** | | | | | | | | | | | | | 0.08 | 0.05 | -0.15* | -0.12 | | | | |
| | | | | | | | | | | | | | (-0.05) | (-0.12) | (-0.08) | (-0.37) | | | | |
| **T_mean_winter** | | | | | | | | | | | | | | | | | 6.6 | 15.65 | 4.59 | 32.78 |
| | | | | | | | | | | | | | | | | | (-4.71) | (-9.79) | (-7.1) | (-30.42) |
| **TxP** | | | | | | | | | | | | | | | | | -0.03 | -0.07** | -0.01 | -0.05 |
| | | | | | | | | | | | | | | | | | (-0.03) | (-0.03) | (-0.02) | (-0.05) |
| **Observations** | 26 | 26 | 25 | 25 | 26 | 26 | 25 | 25 | 26 | 26 | 25 | 25 | 26 | 26 | 25 | 25 | 26 | 26 | 25 | 25 |
| **R2** | 0.12 | 0.8 | 0.89 | 0.57 | 0.69 | 0.87 | 0.93 | 0.61 | 0.65 | 0.86 | 0.81 | 0.57 | 0.72 | 0.87 | 0.94 | 0.61 | 0.71 | 0.9 | 0.93 | 0.63 |
| **Adjusted R2** | 0.09 | 0.79 | 0.89 | 0.56 | 0.66 | 0.86 | 0.92 | 0.57 | 0.62 | 0.85 | 0.8 | 0.53 | 0.68 | 0.85 | 0.93 | 0.56 | 0.66 | 0.88 | 0.91 | 0.56 |
| **Residual Std. Error** | 9.48 | 15.13 | 9.78 | 32.22 | 5.8 | 12.55 | 8.32 | 31.57 | 0.7 | 0.81 | 0.94 | 2.06 | 5.61 | 12.78 | 7.89 | 32.24 | 5.81 | 11.51 | 8.62 | 32.18 |
| **F Statistic** | 3.36* | 96.18*** | 195.83*** | 31.06*** | 25.04*** | 75.83*** | 140.27*** | 17.14*** | 21.00*** | 69.40*** | 47.84*** | 14.79*** | 18.72*** | 48.82*** | 104.98*** | 10.99*** | 12.99*** | 46.61*** | 65.47*** | 8.54*** |
| **Note:** | *p<0.1; **p<0.05; ***p<0.01; std_dv in () | | | | | | | | | | | | | | | | | | | |


**Table A2: results of multiple regression equations tested for representing near-natural streamflow during the reference period in the Winter season**

**Appendix B**

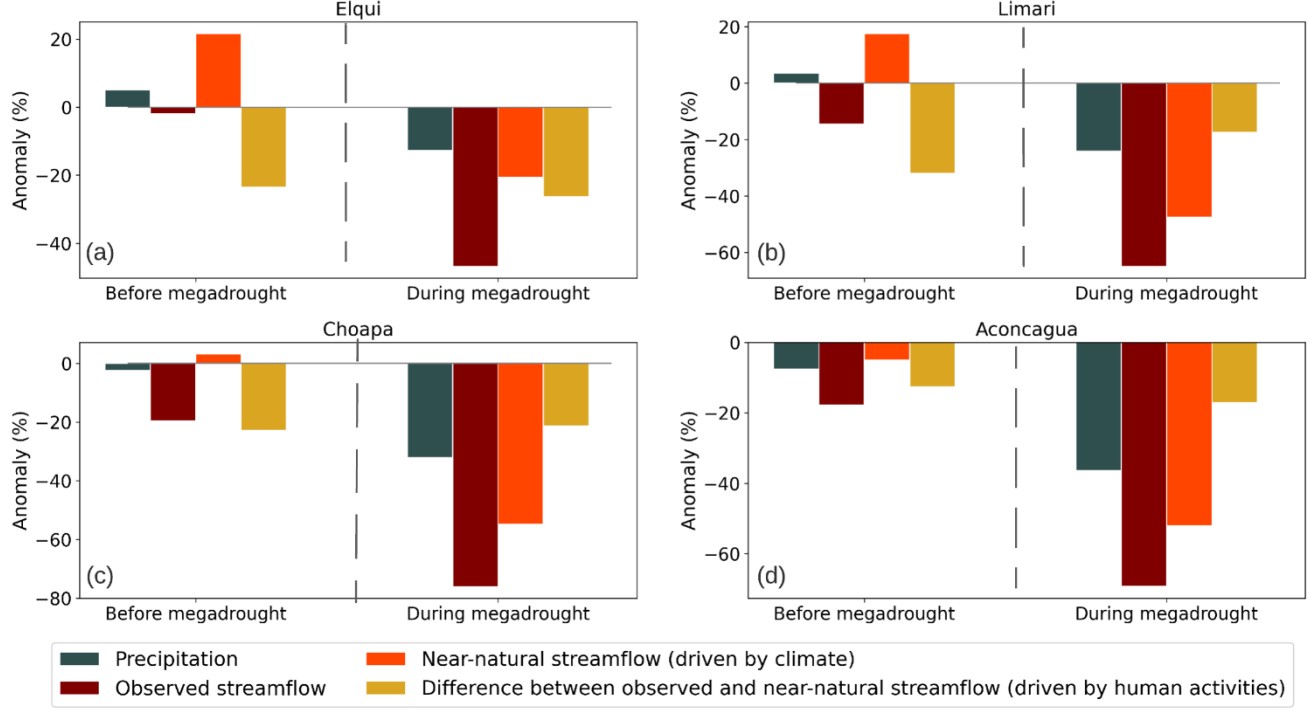


**Figure B1: Anomalies in annual precipitation, observed streamflow, simulated near-natural streamflow and human-induced**
**streamflow change. The anomalies are presented for the evaluation period before and after the megadrought onset (1988-2009 and**
**2010-2020, respectively). For each flux, the anomalies are computed as the percentage difference with respect to their mean values**
**during the low-influence reference period (1960-1988). The graphs show these results for Elqui (a), Limarí (b), Choapa (c), and**
**Aconcagua (d) Basins, respective**

**Appendix C.**

| Basin | Hydrological Drought | Frequency | Duration (seasons) | | | Deficit (mm) | | |
|---|---|---|---|---|---|---|---|---|
| | | | Total season | Max duration | Average duration | Total deficit | Max deficit | Average deficit |

| Basin | | Frequency | Total season | Max duration | Average duration | Total deficit | Max deficit | Average deficit |
|---|---|---|---|---|---|---|---|---|
| Elqui | Near- natural | 4.00 | 6.00 | 2.00 | 1.50 | 18.43 | 10.13 | 4.61 |
| | Observed | 4.00 | 14.00 | 10.00 | 3.50 | 63.15 | 53.09 | 15.79 |
| | diff % | 0.00% | 57.14% | 80.00% | 57.14% | 70.82% | 80.92% | 70.82% |
| Limari | Near- natural | 2.00 | 10.00 | 6.00 | 5.00 | 98.39 | 51.69 | 49.20 |
| | Observed | 3.00 | 13.00 | 8.00 | 4.33 | 94.94 | 65.87 | 31.65 |
| | diff % | 33.33% | 23.08% | 25.00% | -15.38% | -3.64% | 21.53% | -55.46% |
| Choapa | Near- natural | 2.00 | 10.00 | 6.00 | 5.00 | 107.89 | 58.31 | 53.94 |
| | Observed | 4.00 | 14.00 | 8.00 | 3.50 | 212.76 | 123.74 | 53.19 |
| | diff % | 50.00% | 28.57% | 25.00% | -42.86% | 49.29% | 52.88% | -1.42% |
| Aconcagua | Near- natural | 3.00 | 10.00 | 4.00 | 3.33 | 237.65 | 133.27 | 79.22 |
| | Observed | 4.00 | 13.00 | 8.00 | 3.25 | 565.61 | 315.06 | 141.40 |
| | diff % | 25.00% | 23.08% | 50.00% | -2.56% | 57.98% | 57.70% | 43.98% |

**Table C1: drought characteristics for each basin considering the observed and simulated near natural streamflow during the**
**megadrought period (2010-2020). The third row for each basin represents the human influence on drought characteristics as the**
**percentage difference between the observed and the naturalized scenario**

| Basin | Hydrological Drought | Frequency | Duration (seasons) | | | Deficit (mm) | | |
|---|---|---|---|---|---|---|---|---|
| | | | Total season | Max duration | Average duration | Total deficit | Max deficit | Average deficit |
| Elqui | Near- natural | 7.00 | 10.00 | 4.00 | 1.43 | 44.99 | 20.55 | 6.43 |
| | Observed | 6.00 | 10.00 | 4.00 | 1.67 | 30.46 | 12.73 | 5.08 |
| | diff % | -16.67% | 0.00% | 0.00% | 14.29% | -47.69% | -61.49% | -26.59% |
| Limari | Near- natural | 4.00 | 7.00 | 4.00 | 1.75 | 68.65 | 50.78 | 17.16 |
| | Observed | 8.00 | 15.00 | 6.00 | 1.88 | 91.61 | 54.26 | 11.45 |
| | diff % | 50.00% | 53.33% | 33.33% | 6.67% | 25.06% | 6.41% | -49.87% |
| Choapa | Near- natural | 6.00 | 10.00 | 4.00 | 1.67 | 90.04 | 34.36 | 15.01 |
| | Observed | 9.00 | 17.00 | 6.00 | 1.89 | 135.96 | 67.42 | 15.11 |
| | diff % | 33.33% | 41.18% | 33.33% | 11.76% | 33.78% | 49.04% | 0.66% |
| Aconcagua | Near- natural | 5.00 | 9.00 | 2.00 | 1.80 | 180.33 | 67.70 | 36.07 |
| | Observed | 9.00 | 13.00 | 3.00 | 1.44 | 468.17 | 110.86 | 52.02 |
| | diff % | 44.44% | 30.77% | 33.33% | -24.62% | 61.48% | 38.93% | 30.67% |

**Table C2: drought characteristics for each basin considering the observed and simulated near natural streamflow before the mega**
**drought period (1988-2010). The third row for each basin represents the human influence on drought characteristics as the percentage**
**difference between the observed and the naturalized scenario**
