# Peer review of "The influence of human activities on streamflow reductions during the"

_Hydrology and Earth System Sciences, 2023_

## Author Comment (AC1)

**Response to the editor for hess-2023-246**

**Comentado [1]:** poner todo texto agregado en cursiva

We are very grateful for the comments and suggestions from the reviewers that contributed to improving the manuscript. We appreciate the time they spent to evaluate our work. All comments were taken into account and were individually addressed. Note that answers are in blue and sentences added/adjusted in the manuscript are in quotation marks. Lines and figures numbers are to be understood in reference to the first submitted manuscript, so as to be consistent with the reviewer's comments.

**Responses to Reviewer #1.**

**Specific comments**

**R. Abstract:** The abstract is informative, yet quite lengthy and could benefit from concise rephrasing. The initial problem statement (lines 13 -17) and methodology (lines 19-24), spanning several lines, could be condensed to give more emphasis on the results of the study. For instance, the problem statement could be introduced as following: "Since 2010, central Chile has experienced a protracted megadrought. Intensification of the drought has been attributed to the effect of cumulative precipitation deficits linked to catchment memory. Yet, the influence of water extractions on drought intensification is still unclear." Additionally, in line 26, the term 'undisturbed period' is introduced for the first time. For improving clarity, I would suggest explicitly writing that you are referring to a period with low human intervention.

**Authors**: The first paragraph of the abstract has been modified as suggested. The term 'undisturbed' has been replaced with *"low human influence period"* in new line 21 as it is not used elsewhere in the article. The revised abstract now places greater emphasis on the study's results by condensing the problem statement and methodology, aligning with the reviewer's suggestion for conciseness. The abstract now reads as follow in lines 13-30:

"*Since 2010, central Chile has experienced a protracted megadrought with annual precipitation deficits ranging from 25% to 70%. Intensification of the drought has been attributed to the effect of cumulative precipitation deficits linked to catchment memory. Yet, the influence of water extractions on drought intensification is still unclear Central Chile has experienced a protracted megadrought since 2010 (up to date), with annual precipitation deficits ranging from 25% to 70%. Drought propagation has been intensified during this time, with streamflow reductions up to 30% larger than those expected from historical records. This intensification has been attributed to the cumulative effect of precipitation deficits associated with catchment memory in near-natural basins of central Chile. However, the additional effect of water extractions on drought intensification in disturbed basins remains an open challenge. Our study assesses climate and water use effects on streamflow reductions and hydrological droughts during the last three decades in four major agricultural basins, focusing on the ongoing megadrought. We contrast observed streamflow with near-natural simulations, representing discharge that would have occurred without water extractions. In this study, we assess the effects of climate and water use on streamflow reductions and hydrological*

*droughts during the last three decades in four major agricultural basins in central Chile, with a particular focus on the ongoing megadrought. We address this by contrasting streamflow observations with near-natural streamflow simulations representing the discharge that would have occurred without water extractions. Near-natural streamflow estimates are obtained from rainfall-runoff models trained over a reference period with low human intervention (1960-1988). We characterize hydrological droughts driven by precipitation and human activities during the evaluation period (1988-2020) in terms of the frequency, duration and intensity of near-natural and observed seasonal streamflow deficits, respectively.*

*Our results show that before the megadrought onset (1988-2009), streamflow in the four basins was 2 to 20% lower than the streamflow during the low human influence undisturbed period. Between 81 to 100% of these larger deficits were explained by water extractions. During the megadrought (2010-2020), streamflow was reduced in a range of 47 to 76 % among the different basins, compared to the reference period. During this time, the climatic contribution to streamflow reductions increased and water extractions had a lower relative contribution, accounting for 27 to 51% of streamflow reduction. hydrological droughts were characterized in terms of the frequency, duration and intensity of near-natural and observed seasonal streamflow deficits, respectively. During the complete evaluation period, human activities have amplified the propagation of droughts, with more than double the frequency, duration, and intensity of hydrological droughts in some basins, compared to those expected by precipitation deficits only. We conclude that while the primary cause of streamflow reductions during the megadrought has been the lack of precipitation, water uses have not diminished during this time, causing an exacerbation of the hydrological drought conditions and aggravating their impacts on human water consumption, economic activities, and natural ecosystems.* "

**R. Lines 43-46:** this paragraph is a bit disconnected from the first and third paragraphs. I recommend merging it with the first paragraph to enhance overall flow. You could make the text more concise by eliminating the reference to climate change (as it is not the focus of your analysis) and starting to introduce the concept of hydrological drought soon after the first sentence. Or you could connect the first and second paragraphs by introducing a transitional statement like: These alterations in the water cycle may have implications for drought occurrences. If meteorological droughts (..) are mainly controlled by regional climate, hydrological droughts (..) are also influenced by..

**Authors**: As the reviewer suggested the first and second paragraphs were connected by adding the following transitional statement. "*These alterations in the water cycle may have implications for drought occurrences. If …*". Please find this in the lines 39-40

**R. Line 53:** the word "dominated" is not very appropriate and could be replaced with "driven". Also, I think you mean: "anthropic water uses";

**Authors**: The sentence has been modified as suggested. The sentence - in lines 50-51- now reads: "*which has been mainly driven by anthropic activities in the basin*". Anthropic uses were changed for anthropic activities, which include land use changes, construction of water reservoirs, and water extractions between other activities

**R. Line 57-58:** is quite confusing. What do you mean with 'adaptation and mitigation water management plans'?

**Authors**: The adaptation and mitigation part was removed to avoid any confusion. now reads only "*water management plans*." Water management plans also consider strategies to mitigate water risks and to adapt to those impacts that could not be mitigated.

**R. Lines 67-69:** The sentence is quite disconnected from the previous paragraph: above you mentioned the hydrological drought and water scarcity problem in central Chile, while later you give an example of the possible influence of human activities on the propagation of drought. Or you delete: "for example" or you already introduce in the previous paragraph the potential impact that human activities may have on the intensification and propagation of drought;

**Authors:** "For example" was deleted as the reviewer suggest

**R. Line 67:** I would suggest providing the geographical location of the Petorca river basin (as you do for the Aculeo Lake on line 70).

**Authors**: Changed as suggested. Please find it in line 64

**R. Lines 88-89:** It is not very clear: do you mean that annual precipitation concentrates usually during winter (then delete 'a few') or that most of the precipitation during the last TOT years occurred only during a few winters or something else?

**Authors**: this sentence refers to the fact that most of the annual precipitation occurred during the winter concentrated on a few storm events. The sentence was rephrased for better comprehension and now reads "*occurring during the winter season concentrated on a few storm events*". please find this in line 94

**R. Lines 96-97**: While knowing the actual extent of agricultural land cover is valuable, grasping the influence of these areas at the catchment scale becomes challenging without information on the catchment size. I would suggest providing the percentage of the total catchment area covered by agriculture areas. Similarly, it would be good to know the annual water consumption in relation to mean annual flow (particularly in cases of minimal streamflow variation throughout the year).

**Authors**: We appreciate the valuable suggestion. The total catchment area has been added to the text to address this concern. Regarding the annual water consumption comparison to mean annual flow, we want to highlight the significant hydro climatic annual variation in this catchment, leading to substantial variations in mean flow along the upstream-downstream gradient due to water uses. Considering this, providing the mean annual flow might not offer meaningful insights due to the inherent endogenous comparison caused by the catchment's unique characteristics.

**R. Figure 1:** In Panel A, the proxy station (green diamond) in the Limari catchment (light blue) is not visible. Could this be due to the absence of fill-in gaps in the timeseries extracted from the outlet represented by the red diamond? In Panel C, the colors are quite hard to distinguish. The majority of the area is categorized as 'others,' making it less

informative. The agriculture regions are difficult to see, with clearer concentrations observed mainly in Aconcagua and Limari. Moreover, the colors used for urban and mountain landcover are notably similar.

**Authors:** Indeed, the Limari catchment streamflow time series lacks any gap-filling process. To address this concern and avoid any potential confusion, we have included an additional sentence in the caption. It now reads "*Note that in the Limari catchment, no gap-filling process was made, leading to the absence of a predictor gauge station.*" please find this in lines 109-110

In panel C, the colors of the Agriculture and mountain categories changed for better visualization, and the category others were renamed to "*meadows and shrublands*" making it more informative

**R. Line 108:** Monthly mean streamflow? Same applies, for line110: monthly mean or sum precipitation?

**Authors**: The text now clarifies that the data is for total monthly streamflow and total monthly precipitation. The first sentence now reads *"Times series of total monthly streamflow normalized by catchment area (in mm/month)*". please find these in lines 114-116

**R. Figure 2:** In the text under step 2, the use of the plural in indicating the period of low anthropic activities might be confusing. It may appear as if the model was trained in multiple periods, whereas, in reality, only one continuous period of low water consumption was identified. It would be clearer to use 'in the period' instead. Same applies for line 129, where I would write 'in a period'.

**Authors**:  change as suggested

**R. Line 136:** First time you are using the term 'anthropic variables', I would continue using the term 'water uses' for consistency.

**Authors**:  change as suggested

**R. Eq. 2:** Is the rainfall-runoff model applied on an annual or monthly scale? It is good to specify in the text what 't' is. And to change the time scale of your time series do you use the average or the sum? I suggest providing this information in the text.

**Authors**:  The rainfall-runoff model was applied to seasons of six months (winter and summer) scale. Each variable represents the total rainfall/ runoff of the whole season (the sum). This was clarified in the text for better comprehension and now reads "To *account for pluvial and snowmelt runoff generation processes, we implemented seasonal rainfall-runoff models, considering the total runoff and rainfall in the six-month periods defined in Sect 2.2 as dependent and independent variables, respectively*." Please find this in lines 163 -165.

**R. Eq. 2 and Eq. 3:** You are considering a linear relationship between precipitation and runoff, however annual runoff data is generally skewed and not linearly correlated to

precipitation. Several researchers, for example, used a linear rainfall-runoff relationship only after verifying the assumption of a normal distribution of rainfall values and subsequently transforming the runoff data with a Box-Cox transformation to ensure they follow a normal distribution (Saft et al., 2015: The influence of multiyear drought on the annual rainfall-runoff relationship: An Australian perspective, Avanzi et al., 2020: Climate elasticity of evapotranspiration shifts the water balance of Mediterranean climates during multi-year droughts). The decision to employ a linear regression model for capturing the rainfall-runoff relationship should be then justified in the text or explicitly mentioned as a limitation in the analysis within the discussion section. In line 167 -170, you mentioned that multiple regression equations have been tested, however you do not specify in the text which other regression equations have been used. Further clarification on this can provide valuable information.

**Authors**: In response to the reviewer's valuable suggestions, and following the method of Saft et al. (2015), we added a new rainfall-runoff model in which we applied a Box-Cox transformation to the seasonal and annual streamflow. A summary of the performances from the tested models were added into a new Appendix A. The Box-Cox transformed models led to a reduction in model performance across all basins. In consideration of this outcome, we have retained the original linear rainfall-runoff model as presented in the initial draft. In order to discuss methodological limitations, we have added a new section in the discussion to explicitly address and analyze the general limitations of the model associated with the usage of a linear rainfall-runoff relationship, among other methodological issues.

**R. Line 192:** The choice of the threshold for detecting hydrological drought should be justified, since it influences the periods in which near-natural and observed streamflow are compared.

**Authors**:
As stated in the submitted manuscript, we adopted a seasonal threshold of 80% to select drought events, following Rangrecoft et al. (2019). We realize that different threshold values may be tested. In order to assess this effect, we tested a 70% (i.e., more events selected) and a 90% (i.e., less events selected) seasonal threshold and checked that the main results regarding the role of human influence on drought propagation were consistent among these selected values. Since adopting a 70% seasonal threshold leads to the selection of more drought events, the statistical analyses performed over drought characteristics (Table 4) is more robust. Therefore, we adopted this value for presenting the results and computing drought characteristics. This methodological justification is included in the revised Sect. 2.4: *"To identify drought events, thresholds based on percentiles of the flow duration curve are commonly used. For daily or monthly time series, a recommended threshold falls between the 70th and 90th percentile (Rangecroft et al., 2019; Van Loon et al., 2016; Van Loon, 2015). In this study, the 70th percentile of the seasonal streamflow series is adopted to define hydrological droughts. This lowest threshold allows for the selection of more drought events, which makes statistical analysis more robust."* Please find this in lines 202-207

**R. Figure 4:** The terms streamflow and runoff in the caption and figure have been used interchangeably, although different. Is the blue line indicating runoff or annual streamflow? The same occurs in Table 1 (line 234).

**Authors**: The caption and the figure now indicate the term streamflow for the blue line. The same was changed in Table 1.

**R. Table 2:** I found it strange that the mean bias error for Equi and Aconcagua is 0% even though they have a low r-squared (under 70%). Could it be that this is because the model both over- and under-estite duma trring theaining period and so the average bias converges to 0? And how does this low model performance for some catchments in winter affect the results in Figure 7? Could it be that the difference you capture between the near-natural streamflow and the observed streamflow is the error in the model estimation of the near-natural flow instead of the additional water stress resulting from water abstractions? Because with a low r-squared and high mean bias in the regression model for the winter season, it might be that you are capturing the bias in the near-natural streamflow more than the difference between the near-natural streamflow and the observed streamflow.

Further, in the Study Area section (2.1), you also mention that the winter season is the most important for water replenishment since the high concentration in precipitation in that season (Garreaud et al., 2017).

Additionally, you use annual time series on water uses of industry, energy, mining, livestock, and drinking water sectors. But information from the literature on how these uses change over a year (e.g., intensified in summer or winter) can better help interpret the results and might give relevance to a particular season over another.

Some reflections on the above points in the discussion section might be helpful.

**Authors:** Certainly, we appreciate the insightful comments from the reviewer and would like to address their concerns comprehensively. Regarding the Mean Bias Error (MBE) values approaching zero during the calibration period, it's essential to note that these outcomes are expected in the least squared error model regression, indicating an absence of systematic bias in the estimations. The slight deviations from zero are a result of truncating a few negative streamflow estimates to zero. The reviewer rightly draws attention to the potential uncertainties in winter model performance and its impact on the attribution exercise interpretation. Indeed, the naturalized estimations have uncertainties that can mask some of the human influence effects. In order to visualize the potential of this artifact, Figure 6 shows the streamflow estimations with a 95% confidence interval. These plots, in general, show that the lowest values of naturalized streamflows is above the observed time series. An uncertainty analysis was not further developed within the study, although the limitations were discussed in the new discussion section 4.2.3., about methods limitation

Additionally, the reviewer suggests considering the seasonal distribution of water uses to better interpret the results. The seasonal peaks in human water consumption, particularly from agriculture are indeed a key factor influencing water availability during the lowest streamflow season. Since the water consumption was only used to select the low-influence training period

(Sect. 2.3.1), the results of the paper are not affected by this lack of seasonal representation in water consumption. Despite this, the potential implications of water consumption seasonality are mentioned in Sect. 3.1 (*"In contrast, the mean annual streamflow decreased by a range of 14 to 35%. If we examine summer streamflow, when agricultural water consumption is more intense, a reduction of 24 to 46% is observed."*) and further discussed in the new Sect.4.2.3

**R. Figure 6:** I would move Figure 6 in section 3.2 since you already refer to it in that section. You can also think of merging this figure with table 2 (indeed in line 267-268 you mention the r-squared values, which however are not shown in the figure).

**Authors**: In response to the reviewer's suggestion, we have merged Figure 6 with Table 2 and relocated the combined figure to section 3.2, where it is referenced. It's important to note that Figure 6 now exclusively displays the performance in terms of R2, Given the convergence of Mean Bias Error (MBE) to 0 in the least squared error model regression, which could potentially lead to misleading comparisons between models, we have excluded MBE from Figure 6.

**R. Figure 7:** Precipitation and near-natural streamflow have very similar colors, I would differentiate the colors a bit more.

**Authors:** The colors were changed as suggested for better readability, also the units of the y-axis were updated to percentages to be coherent with the text.

**R. Section 4.1:** Would it be possible to compare the results on the impacts of human activities on water availability together with the differences in trends of water users (Figure 5)? Water uses for agriculture and mining differ between catchments and over time with some peaks, lows, and some constant values. Perhaps reflecting on this could provide valuable information for local authorities trying to understand which water sector has the greatest impact on drought intensification.

**Authors:** We computed the difference in water consumption during the low and high human influence periods for each basin (see table below). The assessment reveals distinct patterns among basins, to include this analysis in the discussion the following text has been added in Sect 4.1. "*In general, the basins with the greatest increases in total water consumption during the evaluation period also exhibit higher human influence in the reduction of streamflow. Elqui and Limari exhibited the most significant relative increase in total water consumption, primarily driven by a substantial rise in agriculture consumption from 1989 to 2010, while Choapa almost duplicated its total water consumption during the 2000-2010 decade, due to mining operations. It is noteworthy that agriculture and mining water consumption continued to rise during the megadrought*". Please find this in lines 352-357

| Basin | Surfaces | Drinking | Minery | Agriculture | Total | Period | Total diff |
|-------|----------|----------|--------|-------------|-------|--------|------------|
|       | 0.061    | 0.031    | 0.006  | 0.368       | 0.465 | 1960-1988 |          |
| Elqui | 0.061    | 0.036    | 0.052  | 0.868       | 1.017 | 1989-1999 | 117.91%  |

| | | | | | | Period | % |
|---|---|---|---|---|---|---|---|
| | 0.180 | 0.041 | 0.063 | 1.198 | 1.483 | 2000-2009 | 45.57% |
| | 0.184 | 0.045 | 0.083 | 1.263 | 1.575 | 2010-2020 | 6.19% |
| | 0.000 | 0.036 | 0.002 | 0.329 | 0.367 | 1960-1988 | |
| | 0.000 | 0.045 | 0.021 | 0.519 | 0.585 | 1989-1999 | 58.29% |
| | 0.000 | 0.048 | 0.051 | 0.828 | 0.927 | 2000-2009 | 58.20% |
| Limarí | 0.000 | 0.049 | 0.068 | 0.945 | 1.062 | 2010-2020 | 14.07% |
| | 0.060 | 0.047 | 0.002 | 1.124 | 1.234 | 1960-1988 | |
| | 0.061 | 0.059 | 0.035 | 0.882 | 1.037 | 1989-1999 | -16.06% |
| | 0.101 | 0.066 | 0.813 | 1.130 | 2.110 | 2000-2009 | 102.41% |
| Choapa | 0.135 | 0.075 | 1.212 | 1.179 | 2.601 | 2010-2020 | 23.07% |
| | 0.181 | 0.155 | 0.061 | 2.123 | 2.520 | 1960-1988 | |
| | 0.182 | 0.252 | 0.488 | 2.471 | 3.394 | 1989-1999 | 34.63% |
| | 0.180 | 0.300 | 0.904 | 2.454 | 3.839 | 2000-2009 | 13.11% |
| Aconcagua | 0.184 | 0.346 | 1.072 | 2.646 | 4.248 | 2010-2020 | 10.66% |

**R. Lines 336-337:** maybe good to briefly specify somewhere in the text that groundwater does not contribute to streamflow in the catchment under analysis (if this is the case).

**Authors**: Groundwater does contribute significantly to streamflow in the analyzed catchments. In the Aconcagua catchment, for instance, Crespo et al. (2020) found that groundwater sources contribute up to 46% of streamflow in downstream sections during certain seasons. To explicitly address the reviewer's concern, we have added the following paragraph: "*"Groundwater sources play a crucial role in streamflow within the study area's basins (Crespo et al., 2020; Taucare et al., 2020; Alvarez-Garreton et al., 2021). The impacts of declining groundwater levels include disruptions in water access in rural areas and the potential disconnection between surface and underground water sources, leading to a decrease in soil moisture conditions (agricultural drought) and the desiccation of rivers and lakes (Duran-Llacer et al., 2022; Muñoz et al., 2020). This exacerbates hydrological drought, delaying the recovery of catchments from drought episodes. Also, agricultural shifts from surface to groundwater sources, intended to alleviate megadrought impacts, leads to inelastic water consumption rates. This increases the water stress levels within a basin and reduces groundwater reservoirs, which could ultimately lead to an absolute day zero (Alvarez-Garreton et al., 2023)"*. Please find this in lines 364-371

**R. Lines 357-372:** There is a bit of disconnection between the previous paragraph and this one. Although you start the paragraph by referring to the intensification of streamflow deficits, you then discuss drought impacts (357-363) to then start to talk about human

activities (Line 363). I would actually remove the part from ''Although..''(line 357) to "US$56 million during 2010-2020" (line 363) or move it below.

**Authors:** Section 4.2 of the discussion has been revised to address your feedback. The discussion of drought impacts has been removed from the second paragraph. Additionally, a final paragraph has been added to provide a comprehensive overview that highlights the broader impacts of hydrological drought, consequences of maladaptation, and the persisting challenges. The revised paragraph now reads as follows:

"*The resulting precipitation deficit and human activities, including human-induced maladaptation processes, have broad, complex, and exacerbated impacts on society and ecosystems. For example, agricultural practices not only worsen water scarcity but also contribute to soil erosion and sediment transport, further degrading ecosystem health. The intensified streamflow deficits disrupt watercourses and contribute to significant tree mortality (Miranda et al., 2020). Additionally, thousands of people have lost access to domestic water services (Muñoz et al., 2020), leading to significant spending on water cistern trucks. There is still a gap in understanding how human activities contribute to catchment vulnerability to hydrological droughts and how the connections within the hydrological cycle are being assessed in drought management (Anne F. Van Loon et al., 2016). This underscores the urgent need for sustainable water management strategies in the face of changing hydrological conditions.*" Please find this in lines 402-411

**R. Grammar check:** Line 56: ''to assess catchments' vulnerability''; Line 60: ''we focus on central Chile'';  Line 71: 'by using' instead of 'and used' (to reduce the use of 'end'); Line 96: land cover (without s); Line 176: 'compared'; Line 199: 'defined' (consistent with the past tense form used for the method section).

**Authors:** Change as suggested

**Responses to Reviewer #2.**
**General comments:**

**R. Section 3.1**. Why did the authors not exploit water use data only to identify periods with low/high human influences? Provided good quality (some more details on them would be helpful anyway, given the common challenges around collecting this kind of data), I think these could give direct information on change points in the different basins due to human intervention.

**Authors**: The reviewer has a good point. One option to identify low-intervention periods is to use water use data to identify change points and then check if those points are consistent with changes in streamflow. The option we took for this selection was the inverse: to identify change points in the streamflow time series and then corroborate that these points were supported by water use changes.

It's crucial to note that the complexity of water transfers within the study basin makes sub-catchment consumption data potentially non-representative of the broader impacts on streamflow. Despite the valuable information water use data could provide, uncertainties exist in their ability to accurately capture the diverse manifestations of anthropogenic intervention in the basins. Challenges such inter-basin water transfers and land cover changes contribute

to this uncertainty. Therefore, we opted for a comprehensive approach, considering multiple variables including streamflow, precipitation, and anthropic variables, to ensure a robust identification of breakpoints in streamflow that were not predominantly influenced by climatic variations.

**R. Section 3.2.** I appreciated that the authors investigated also potential climate-driven non-stationarities on Q during the multi-year drought, given their evidence in Chilean catchments (Alvarez-Garreton et al., 2021) and their relevance for hydrological modelling (e.g., Saft et al., 2016). I assume runoff ratios in Table 3 were computed on naturalized Q, given the Section in which they appear. If so, how can we infer that human activities are responsible for the intensification of the Q reductions (L254-256)? Furthermore, are the upstream-downstream catchments fully comparable? Could you please provide more information on their catchment attributes? In some cases (e.g., Choapa), I wonder if additional factors - other than human influences only - may play a role in the different behaviour of the catchments, given the significant difference in catchment area. Why not computing shifts in rainfall-runoff relationship (e.g., Saft et al., 2015, Alvarez-Garreton et al., 2021) on naturalized Q for downstream basins? This could provide a clue of potential climate-driven non-stationarities in the basins during the megadrought. Furthermore, if such non-stationarities happen, how do you ensure that the linear model for the naturalization can properly reproduce Q? I recommend adding at least some discussion on this latter point. Concerning the choice of the naturalization approach, I also wonder why upstream-downstream comparison (e.g., Van Loon et al., 2022 and references therein) was not chosen, provided comparability of the upstream/downstream catchments.

**Authors**: First, it is important to clarify that the rainfall-runoff ratios of Table 3 correspond to observed streamflow. This was clarified in the method section when the rainfall-runoff ratios are defined "*we compared the observed rainfall-runoff ratios (mean annual observed streamflow runoff normalized by mean annual precipitation*". Please find this in lines 186-187. Given that we are considering observed streamflow, the intensification in streamflow reductions is attributed to the combination of human activities, natural hydrological processes, and the potential effects of non-stationarity catchment response. Since the upper catchment sections have a lower human influence than the downstream sections, the larger streamflow decrease during the MD (compared to the previous period) in these sections may be mostly attributed to non-stationarity in basin response during protracted droughts (consistent with Saft et al., 2015, Alvarez-Garreton et al., 2021). However, the downstream sections feature an even larger streamflow reduction during the MD, compared to the reduction in the upper sections, which are related to the added effect of human activities, which have maintained water consumption despite the reduced water availability (Fig. 5). We added a discussion of this in the new method limitations discussion Sect. 4.2.3

An upstream-downstream comparison approach to analyze the influence of human activities was not adopted since the upper sections of the basins are not fully natural catchments. Although they have a lower water consumption than downstream sections, they are still influenced by human activities and thus cannot be used as a proxy for naturalized conditions. For example, a large mine was installed in the upper part of the Aconcagua basin in the 1980s (Fig. 5), drastically modifying the streamflow in the Aconcagua river. We overcome this by adopting a linear model to get naturalized streamflow estimates.

**R. Section 3.3.** I would suggest focusing on a 10-year period, instead of 1988-2009, to ensure the same record length for the computation of anomalies before and during the megadrought.

**Authors:** We appreciate the suggestion of the reviewer to focus on a 10-year period for consistency in computing anomalies before and during the megadrought. However, our primary objective is to highlight the differences in human influence on drought propagation before and during the megadrought. Further, separating the evaluation period into three decades would lead to very few drought events for selection and statistical analysis (Sect. 2.4).

**R. Section 3.4.** The fact that most streamflow drought events after 2010 in Figure 8 appear mostly as human-induced may sound a bit contrasting the result of decreasing human contribution to Q reduction during the megadrought (Figure 7). I suggest showing in Table 4 the relative difference in drought characteristics due to human influences (e.g., Van Loon et al., 2022), before and during the megadrought to provide additional insights on this. An additional event-scale analysis on the role of human activities in aggravating/inducing streamflow droughts could also provide novel understandings on the role of human influences during prolonged meteorological droughts (L370-372).

**Authors:** Taking into consideration the reviewer's comment we have quantified the influence of human activities over observed hydrological drought by calculating the relative difference in each drought characteristic (DC). This was explicitly introduced in the method section 2.4, which first paragraph in lines 191-196 now reads: "*To quantify the impact of human activities on hydrological droughts (schematized in Fig. 3), we compared the characteristics of the observed and the near-natural streamflow deficits during drought events, including their frequency (number of drought events), duration (average, maximum and total), and intensity (i.e., deficit of volume) across the evaluation period. In this way, we assess the influence of human activities over observed hydrological drought by calculating the relative difference in each drought characteristic (DC) in the observed and near natural scenario as indicated eq. 5*

$$\Delta DC_{human} = \frac{(DC_{obs} - \widehat{DC_{nn}})}{DC_{obs}}*100 \quad (5)"$$

Then, in section 3.4 of the results, a third row for each basin was introduced in Table 4 (now table 3), illustrating human influence as the percentage difference between drought characteristics in the observed and the naturalized scenario for each respective basin. A comparison of the change of human influence over the drought characteristics between pre and post-megadrought periods was also presented in the text. The following paragraphs have been added to sect 3.4 of the results:

"*The human influence over hydrological drought quantified as the relative difference between the observed and near-natural scenario, varies between the different drought characteristics, but is almost always positive, showing that the human activity aggravates downstream drought characteristics, being responsible for 25 -45% of the total drought events and a 17 -62% of the total streamflow deficit under the drought threshold in the hole period. the negative difference in average duration or average deficit in Limari and Aconcagua it's due to the presence of a*

*greater number of short events. but considering that the total number of events is larger in the observed scenario, this is not indicative of an alleviation of the drought.*

*When analyzing drought characteristics separately before the megadrought (1988-2009) and during the megadrought (2010-2020), Elqui exhibited few human impacts before, but during the megadrought, it notably increased, contributing to 57% of total drought events and nearly 70% of the observed deficit. In contrast, Limari, Choapa, and Aconcagua show relatively stable human contributions to drought characteristics, with a decrease in human contribution to total events (close to 25%), a decrease in Limari's human contribution to the total deficit and a slight increase in Choapa's contribution to the total deficit during the megadrought".* Please find this in lines 327-338

**R.** Why was the drought analysis performed at seasonal (6-month) scale and not monthly, as frequently done? A monthly scale is usually relevant for water management and monthly data are available here (L108). Furthermore, why not using near-natural simulated streamflow for the whole period for the identification of the drought threshold (L201)?

**Authors:** The decision to conduct the drought analysis at a seasonal (6-month) scale, rather than a monthly scale, was driven by the unavailability of monthly information for the modeled streamflows. Monthly flows are inherently influenced by various processes and phenomena throughout the hydrological year, posing significant challenges for accurate modeling with a regression method. Consequently, we opted for a seasonal scale to facilitate a meaningful comparison of drought events between observed and simulated scenarios and keep coherence with sect 3.3.

Addressing the question regarding the use of near-natural simulated streamflow for identifying the drought threshold, our approach aligns with the methodology of Rangecroft et al. (2019). Our objective is to assess human influence by comparing it with the 'natural' situation, encompassing climate-induced changes while excluding human impacts.
In response to the reviewer's suggestion, we revised our methodology by adopting the near-natural simulated streamflow for the entire study period when identifying the drought threshold. Table 4 and figure 8 have changed due to the new threshold identification method.

**R.** While I appreciated the schemes to illustrate the methods, I believe figures could be more informative in general, and Figure 2 itself as well, since it does not fully convey the two main analyses on the effect of human activities on seasonal/annual Q and on streamflow droughts. To this end, I suggest for instance highlighting the megadrought period (here and elsewhere, to help in the readability of the time series as well). Moreover, Figure 5 is hard to read, due to the different ranges of variability for the time series. Maybe a log-scale or two different y scales could help? A common x-scale for the different subplots in Figure 6 would also facilitate comparison among them. I would also enjoy the same units in the figures and text (e.g., percentages in Figure 7, since anomalies are then

presented in percentage at L279fff) to make figures easier to grasp, as well as the same y scales in case of multiple subplots. Finally, please make sure figures and tables are mentioned in the text in the same order as they appear, which is not the case for Figure 8 and Table 4 at the moment.

**Authors:** We have carefully considered your suggestions and made the following adjustments:

Figure 2: While we appreciate your suggestion to highlight the megadrought period, Figure 2 remains unchanged. The specific hydrological drought analysis is conceptualized in Figure 3. In contrast, figure 2 aims to provide a general overview rather than focusing on the megadrought period.

Figure 5: We experimented with both log-scale and two different y-scales, but the vast differences in magnitude for each variable in every basin posed challenges in selecting suitable variables for representation on separate axes. Consequently, we opted to retain the original figure to ensure a nuanced interpretation of variable magnitudes.

Figure 6: To facilitate comparison, all subplots in Figure 6 now share the same x-axis, aligning with your suggestion.

Unit Consistency: We have revised Figure 7 to present anomalies in percentages, aligning with the text for coherence.

Figure and Table Order: We have meticulously reviewed and edited the text to ensure that figures and tables are mentioned in the correct order as they appear in the manuscript.

**specific comments:**

**R. L81:** what "for the same period" refers to is not completely clear, since two periods are mentioned in the sentence before.

**Authors**: the phrase was changed to "*for both periods*", which is what has been done. please find this in line 84

**R. L90**: how are potential changes in glacier cover overtime taken into account in the analysis? If they are not, a brief discussion on this point would be helpful.

**Author**s: We did not include glacier contribution to runoff in our study, and a discussion of this limitation was added to the new method limitation Sect. 4.2.3.

The reason for not including glaciers is because not all the study basins have mountain glaciers. However, the Aconcagua basin receives a significant contribution from glaciers (Crespo et al., 2020), which could influence the interpretation of our results. If the peak water (maximum glacier runoff considering volume and melt rate) has been reached, diminishing melting rates may lead to a reduction in natural streamflow, which can lead to an overestimation of human impacts in our approach. Additionally, in drier summers, glacier contributions tend to rise, potentially mitigating human-induced impacts and leading to a sub-estimation of the real human impacts over streamflow. The intricate dynamics of peak water and seasonal glacier runoff contribution basin merit further investigation, notably due to the scarcity of specific historical time series of glacier runoff contribution and glacier peak-water studies in this region. Ayala et al. (2020) observed in the Maipo River basin (neighbor southern

basin) that glacier contributions to streamflow during hyper-drought in 1968-1969 surpassed the melting contribution during the present megadrought, suggesting a potential pre-megadrought peak water scenario in the Aconcagua basin.

**R. L111:** where do basin polygons come from?

**Authors**: Basin polygons were available on the CAMELS-Cl dataset. This information was added to the text at the beginning of the paragraph, that now reads: *"catchment limits and times series of total monthly streamflow normalized by catchment area (in mm/month) were obtained from the CAMELS-CL dataset (Alvarez-Garreton et al., 2018; available at: https://camels.cr2.cl/)"*. Please find these in lines 114-115

**R. L166**: how is the uncertainty of the regression model quantified? Is it then used to derive the human influence later other than being visualized in Figure 6?

**Authors:** The uncertainty of the regression model comes from the adjustment of the regression to the data in the training period and is represented by the 95% confidence interval of the streamflow estimation coming from the regression parameters estimation. this was clarified by rephrasing the sentence as follows: "$\varepsilon$ *represents the uncertainty of the regression model coming from the parameters estimation*." Please find this in lines 174-175

Subsequently, this uncertainty is utilized for analytical purposes. Specifically, to ensure that the mean observed streamflow lies outside the mean 95% confidence interval simulated by the model during the evaluation period (Fig. 6). This stringent criterion helps identify significant human impacts in the evaluation period, highlighting instances where the observed streamflow deviates significantly from what is predicted by the near-natural model.

**R. L169-170:** could you please add these results in an appendix? This could also add more confidence to the model.

**Authors:** Appendix A has been added to present the results of various models that were tested. Indeed, most models were found to be only dependent on precipitation (P), and no other model exhibited significance for all four basins simultaneously.

**R. Figure 3:** I would plot a variable drought threshold to make the figure more informative, since this kind of threshold is then actually used.

**Authors**: change as suggested

**R. L192**: a sensitivity analysis on the influence of this threshold on the conclusions would be valuable, similarly to what was done for the statistical tests to identify periods with high/low human influences.

We agree with the reviewer that a sensitivity analysis on the adopted threshold is a crucial aspect of drought impact assessment. We acknowledge the importance of considering various factors in this analysis, such as fixed or variable thresholds, low influence periods, the entire dataset, and different magnitude percentiles (e.g., 70th, 80th, or 90th percentiles). However, for a detailed exploration of these considerations, we refer to the work of Rangrecoft et al. (2019), which delves into these aspects comprehensively.

Although a sensitivity analysis is out of the scope of this paper, we tested different threshold values following also a comment from the first reviewer. In addition to the 80th percentiles, we

tested a 70% (i.e., more events selected) and a 90% (i.e., fewer events selected) seasonal threshold and checked that the main results regarding the role of human influence on drought propagation were consistent among these selected values. Since adopting a 70% seasonal threshold leads to the selection of more drought events, the statistical analyses performed over drought characteristics (Table 4) is more robust. Therefore, we adopted this value for presenting the results and computing drought characteristics. This methodological justification is included in the revised Sect. 2.4: *"To identify drought events, thresholds based on percentiles of the flow duration curve are commonly used. For daily or monthly time series, a recommended threshold falls between the 70th and 90th percentile (Rangecroft et al., 2019; Van Loon et al., 2016; Van Loon, 2015). In this study, the 70th percentile of the seasonal streamflow series is adopted to define hydrological droughts. This lowest threshold allows for the selection of more drought events, which makes statistical analysis more robust.* Please find this in lines 202-207

**R. L271:** how were anomalies computed? Please clarify in the Methods.

Authors: This information was available in the caption of Figure 7. To clarify any misleading, In the Methods section 2.3.2 the following sentence was added "*In the results of the attribution exercise (Sect 3.3; Fig 7), hydro climatic variables are depicted as anomalies, computed as the percentage difference from their mean values during the reference period (1960-1988)*". Please find this in lines 176-177

**R. L271:** "observed and near-natural simulated streamflow" -> "observed and near-natural simulated SUMMER streamflow"?

**Authors**: change as suggested

**R. L273:** "Appendix A (Fig. A1)" -> redundant cross-reference to the figure

**Authors**: corrected, now it reads only 'Appendix A"

**R.** I suggest a careful check of the language throughout the manuscript to avoid inconsistencies in verb tenses and forms (e.g., both past and present in the same sentence, L287-288), and terminology (e.g., STARS/Rodionov test, anthropogenic/anthropic, streamflow/flow/runoff used rather interchangeably which generates some confusion), as well as potential typos, also in the legends of the figures (e.g., "this suggest", L332, "treshold" in Figure 3).

**Authors**: change or standardized as suggested

**R.** I think some re-organization among the sections would help the reading flow (e.g., I suggest moving L210-211 to the Methods and L212-214 to the Discussion, and so on).

**Authors:** We appreciate the reviewer's input. To enhance the reading flow, we have moved lines 210-211 to the Methods section 2.3.1 "*In order to select periods with minimal human activities, it is important to identify breakpoints in the streamflow time series that are not primarily explained by climate shifts*". Please find these in lines 148-149. However, regarding the suggestion to move lines 212-214 to the Discussion section, we have decided to retain this

content in the Results section. We believe that these lines contribute significantly to provide a comprehensive narrative on the selection of periods, offering essential context for the interpretation of the results. If further adjustments are recommended, we are open to considering them.

**R. L393**, please check the first link, since I got a "404 Error" when trying to access the website.

Authors: The link was checked, and it is working fine

**Responses to Reviewer #3.**
**General comments**

**R. GC1.** Please provide a more thorough definition of hydrological drought in the introduction as per the methods adopted.

Authors: In response to the reviewer's suggestion, we have enhanced the introduction to include a more comprehensive definition of drought, aligning with the methods employed in our study. Specifically, the following sentences have been added:
"*Drought is defined as a deficit of water relative to normal conditions and can be identified in various components of the hydrological cycle.*" please find this in lines 39-40
"*Hydrological droughts are identified by streamflow deficit using a threshold determined from the near-natural scenario, allowing for better identification of human impacts (Van Loon, 2016).*" please find this in lines 87-88

**R. GC2.** It would be good to define rainfall/climate elasticity and streamflow a little better and perhaps include a reference or two. It will focus more on why it is important to try and include close to natural-periods/catchments to compare to. Water extractions are bound to affect hydrological drought but by how much will be influenced by how elastic or inelastic the relationship between rainfall and streamflow is.

Authors: To address the importance of rainfall sensitivity to precipitation and including close-to-natural stream flows for comparative analysis, we have added a new paragraph in the introduction that reads:

"*A previous study demonstrated that basins with extensive human intervention within this region exhibit relatively lower runoff sensitivities to precipitation compared to less disturbed ones (Alvarez-Garreton et al., 2018). In that study, the authors attributed this phenomenon to the alteration of runoff generation mechanisms associated with water withdrawals and reservoirs. In highly intervened basins, it is essential to simulate near-natural streamflow to disentangle the relative effects of climate and human intervention on drought propagation (Chiew et al., 2006).*," Please find this in lines 77-81

**R. GC3.** In your discussion, you make the point that extraction is inelastic, and you allude to why this can be the case in the conclusion i.e. farmers being issued with additional water rights. I think that it would be important to discuss this in a bit more depth, particularly

around how a potential switch is being made toward groundwater. Importantly, discuss how the current policies might tip the system into the next drought category – agricultural drought.

**Authors:** To explicitly address the reviewer's concern, we have added the following paragraph in the discussion after line 337:

*"Groundwater sources play a crucial role in streamflow within the study area's basins (Crespo et al., 2020; Taucare et al., 2020; Alvarez-Garreton et al., 2021). The impacts of declining groundwater levels include disruptions in water access in rural areas and the potential disconnection between surface and underground water sources, leading to a decrease in soil moisture conditions (agricultural drought) and the desiccation of rivers and lakes (Duran-Llacer et al., 2022; Muñoz et al., 2020). This exacerbates hydrological drought, delaying the recovery of catchments from drought episodes. Also, agricultural shifts from surface to groundwater sources, intended to alleviate megadrought impacts, leads to inelastic water consumption rates. This increases the water stress levels within a basin and reduces groundwater reservoirs, which could ultimately lead to an absolute day zero (Alvarez-Garreton et al., 2023)".* Please find this in lines 364-371

And then, at the end of section 4.2 of the discussion, we added:
*"It is crucial to emphasize that the current water management policy inadequately addresses the physical constraints of groundwater availability, contributing to its failure in preventing water stress conditions (Alvarez-Garreton et al., 2023). Urgent modifications are imperative to ensure sustainable water use and prevent the exacerbation of water stress conditions in the region".* Please find this in lines 409-412

**R. GC4.** Please further clarify some of the consequences of maladaptation. Perhaps you can add thoughts to "there is still a gap in understanding how human activities contribute to the basin's vulnerability to drought."

**Authors:** The section 4.2 of discussions was edited. Droughts impacts were deleted from the second paragraph and the following final paragraph that highlights hydrological drought impacts, maladaptation consequences and remaining challenges was added. "*The resulting precipitation deficit and human activities including human-induced maladaptation processes have broad, complex and exacerbated impacts on society and ecosystems. For example, Agricultural practices not only worsen water scarcity but also contribute to soil erosion and sediment transport, further degrading ecosystem health. The intensified streamflow deficits disrupt watercourses and contribute to significant tree mortality (Miranda et al., 2020). Additionally, thousands of people have lost access to domestic water services (Muñoz et al., 2020), leading to a huge spending on water cistern trucks. There is still a gap in understanding how human activities contribute to catchment vulnerability to hydrological droughts and how the connections within the hydrological cycle are being assessed in drought management (Anne F. Van Loon et al., 2016). This underscores the urgent need for sustainable water management strategies in the face of changing hydrological conditions.*" Please find this in lines 402-411

**Specific comments**

**R. Line 52:** Do you mean "no" significant rainfall deficits?

**Authors:** In effect there was a grammar error. "not" was changed for "no"

**R. Line 168:** perhaps use, instead of – i.e. forms and variables, including evapotranspiration and temperature, were tested. This is also slightly vague as none of the equations list T and ET in their current form. Are you suggesting that all models ended up being only dependent on P?

**Authors:** The text was edited as suggested, removing the "–" and ensuring clarity in the representation. Additionally, Appendix A has been added to present the results of various models that were tested. Indeed, most models were found to be only dependent on precipitation (P), and no other model exhibited significance for all four basins simultaneously.

**R. Lines 221 -228:** Totally understand that much of the information that is provided is based on local experience and while some events are very defined such as the construction of a dam, it would be good to have references to point to changing land use in the region if it exists.

**Authors:** The inclusion of references to changing land use patterns is indeed a valuable suggestion. While we acknowledge the importance of referencing specific events or trends related to changes in land use, we encountered practical constraints that hindered the comprehensive analysis of land use changes in the region.

Initially, our intention was to utilize time series data in elaboration for a thorough examination of land use dynamics. However, due to the unavailability of timely data, this aspect of the study could not be pursued in-depth. Future research endeavors may benefit from a more detailed investigation into land use changes when data availability and time constraints permit.

**R. Lines 293-296:** Agricultural water demand will increase in drought – perhaps you can tease out this paradox a little more.

Authors:In the discussion, we delve into the observed increase in water consumption, particularly in the agricultural sector, during the megadrought. As highlighted "In the Aconcagua basin, the total water consumption increased during the megadrought, while in the other three basins, the human-induced streamflow reduction expressed as mm is slightly smaller during the megadrought, compared to the period prior to the megadrought (Fig. 7). This finding could be explained by an initial reduction in agricultural water consumption during the first years of the megadrought, which was later reversed (Fig. 5a) by higher extractions of groundwater sources in the subsequent years (Taucare et al., 2020; Duran-Llacer et al., 2020)."

To reinforce this explanation and prevent any potential misinterpretation of the results presented in lines 293-296, we have explicitly stated, "*This apparent contradiction may be attributed to the Aconcagua's increased total water consumption during the megadrought, led by intensified agricultural water demand (fig 5a) during drought conditions*". Please find this in lines 306-307

**R. Line 332:** perhaps make clear that total consumption is the sum of surface and groundwater. This makes the link with lines 335-337 clearer.

**Authors**: change as suggested. now the sentences reads *"This suggests that total water consumption from surface and groundwater sources has been inelastic to the surface water deficits"*. Please find this in line 358

---

## Author Response (AR2)

**Response to the editor for hess-2023-246**

We are very grateful for the comments and suggestions from the reviewers that contributed to improving the manuscript. We appreciate the time they spent to evaluate our work. All comments were taken into account and were individually addressed. Note that answers are in blue and sentences added/adjusted in the manuscript are in quotation marks. Lines and figures numbers are to be understood in reference to the first submitted manuscript, so as to be consistent with the reviewer's comments.

**Responses to reviewer #2**

**R.** 1. Reply to my general comments 1 and 5: I thank the authors for their explanations to my questions regarding specific methodological choices (i.e., use of multiple variables in the breakpoint analysis and seasonal resolution). I would suggest the authors briefly add their arguments also in the manuscript, to clarify these points to all readers as well.

**Authors:** We appreciate the suggestion of the reviewer and in response, we have incorporated methodological justifications into the methods section. Addressing general comment 1 concerning the selection of low-influence reference periods, we have revised the sentence in line 152 from *"By employing this approach, we ensure the selection of streamflow breakpoints that are not predominantly influenced by climatic variations"* to *"breaking points in both streamflow and human activities time series, while ensuring the absence of discernible precipitation shifts. We analysed multiple variables instead of using only water use data to achieve a more robust selection of the training period. This reduces the effects of inter-basin water transfers and land cover changes, which may obscure the ability of water use data to accurately capture the magnitude of anthropogenic intervention in the basins."*

Regarding general comment 5, we have addressed this by adding the following text in lines 204-208 of subsection 2.4 (Hydrological drought events characterization): *"In this way, we assessed the influence of human activities over observed hydrological droughts by calculating the relative difference in each drought characteristic (DC) in the observed and near natural scenario. To keep consistency with the attribution methodology (Sect. 2.3), drought events were characterised at a seasonal scale, as indicated in Eq. 5."*

**R.** 2. Reply to my general comment 4: I am glad to see that the authors agreed on the hint of the event-scale analysis to get novel insights on human influences on hydrological drought characteristics during prolonged droughts, and with their additions on that in Sections 2.4 and 3.4. In my opinion, adding some discussion in Section 4 on the results of this analysis and their consistency with other findings – from either previous literature and the current study - would also be nice. Regarding the new Table 3, I recommend to double-check the headings, as I assume they should be the same as in Tables C1–2, and revising the number of digits reported (here and elsewhere, to make the numbers reported in the text easy to grasp from the tables as well).

**Authors:** In response to the suggestion of adding a discussion in Section 4 on the results of this analysis and their alignment with existing literature, we have incorporated relevant discussions in Section 4.1. Specifically, lines 382-396 now emphasize the following:

*"Despite a general decrease in the impact of human influence on streamflow reductions between the pre-megadrought and megadrought periods, the Limari, Choapa, and Aconcagua basins show a relatively stable human contribution to drought characteristics before and during the megadrought, while the Elqui basin experiences a notable increase in human contribution. These observations highlight two key insights. First, they suggest that human activities have a greater influence on drought conditions and characteristics than the solely relative impact of human activities on total streamflow reductions. In the context of meteorological drought, increased and inelastic human water demand exacerbates streamflow reductions, causing them to exceed hydrological drought thresholds in terms of both magnitude and frequency. Second, the increase in human contribution to drought characteristics in the Elqui basin leads it to similar patterns of hydrological drought conditions than the other basins, despite having lower precipitation deficits. This suggests that the role of human water demands is particularly relevant in semi-arid basins with limited precipitation and high interannual variability in terms of precipitation regime, such as Elqui. Consequently, highly intervened basins in semi-arid regions are more susceptible to experiencing severe hydrological droughts during periods of precipitation deficits. These findings align with the observations of Huang (2016), who highlighted that sustainable agricultural development is threatened in arid and semi-arid regions due to limited available water resources, and with Saft et al. (2016), who demonstrated that aridity is a crucial factor influencing streamflow sensitivity to interdecadal climate variability."*

Regarding the headings of tables and number of digits reported. We have revised Tables 3, C1, and C2 for consistency in headings. Additionally, we have ensured uniformity in the number of digits reported across all tables, establishing two digits after the decimal point for clarity and ease of comprehension.

**R.** 3. Reply to my specific comment 2: I assume the second part of the reply is intended to be the addition in Section 4.2.3, even though it is not formatted as the other additions and I cannot see it in the abovementioned section. I encourage the authors to double-check this point, if they really meant to add this paragraph to the discussion.

**Authors:** Thank you for bringing this to our attention. Upon reviewing our decision, we have indeed opted not to include the paragraph in question in Section 4.2.3. This decision was made after thorough consideration, as the topic of glaciers was not previously addressed or problematized in the text. Therefore, it was considered more appropriate to maintain coherence within the discussion section by refraining from introducing new topics that had not been adequately introduced earlier in the manuscript.

**R.** 4. I think that the novelty and relevance of the work for the international community could be stressed better throughout the manuscript (see also the introduction of my first review).

**Authors:** In response to the reviewer's suggestion we have revised the conclusions to better highlight the significance of our findings. The last paragraph of the conclusion now reads as follows:

*"This paper demonstrates that during long and persistent dry periods, human activities in basins in central Chile have intensified drought propagation, by increasing both the intensity and the duration of hydrological droughts. This highlights the importance of understanding the impacts of human activities on drought propagation, and to consider such evidence in water management policies. In particular, to prevent implementing maladaptive measures, the feedback loop between water usage, human activities, and the hydrological system should be considered in the adaptation strategies. These considerations are particularly important not only in Chile but also in other regions worldwide, where the dry signal is consistent and expected to persist."*